# DynaMS: Dyanmic Margin Selection for Efficient Deep Learning

**Jiaxing Wang**[1], **Yong Li** [*][1], **Jingwei Zhuo**[1], **Xupeng Shi**[2], **Weizhong Zhang**[3],
**Lixing Gong**[1], **Tong Tao**[1], **Pengzhang Liu**[1], **Yongjun Bao**[1], **Weipeng Yan**[1]
[1]JD.com      [2]Northeastern University      [3]Fudan University
{wangjiaxing41,liyong5,zhuojingwei1,gonglixing}@jd.com
{taotong,liupengzhang,baoyongjun,Paul.yan}@jd.com
shi.xup@northeastern.edu, zhangweizhongzju@gmail.com

## Abstract

The great success of deep learning is largely driven by training over-parameterized models on massive datasets. To avoid excessive computation, extracting and training only on the most informative subset is drawing increasing attention. Nevertheless, it is still an open question how to select such a subset on which the model trained generalizes on par with the full data. In this paper, we propose dynamic margin selection (DynaMS). DynaMS leverages the distance from candidate samples to the classification boundary to construct the subset, and the subset is dynamically updated during model training. We show that DynaMS converges with large probability, and for the first time show both in theory and practice that dynamically updating the subset can result in better generalization. To reduce the additional computation incurred by the selection, a light parameter sharing proxy (PSP) is designed. PSP is able to faithfully evaluate instances following the underlying model, which is necessary for dynamic selection. Extensive analysis and experiments demonstrate the superiority of the proposed approach in data selection against many state-of-the-art counterparts on benchmark datasets.

## 1 Introduction

Deep learning has achieved great success owing in part to the availability of huge amounts of data. Learning with such massive data, however, requires clusters of GPUs, special accelerators, and excessive training time. Recent works suggest that eliminating non-essential data presents promising opportunities for efficiency. It is found that a small portion of training samples [1] contributes a majority of the loss (Katharopoulos & Fleuret, 2018; Jiang et al., 2019), so redundant samples can be left out without sacrificing much performance. Besides, the power law nature (Hestness et al., 2017; Kaplan et al., 2020) of model performance with respect to the data volume indicates that loss incurred by data selection can be tiny when the dataset is sufficiently large. In this sense, selecting only the most informative samples can result in better trade-off between efficiency and accuracy.

The first and foremost question for data selection is about the ***selection strategy***. That is, how to efficiently pick training instances that benefit model training most. Various principles have been proposed, including picking samples that incur larger loss or gradient norm (Paul et al., 2021; Coleman et al., 2020), selecting those most likely to be forgotten during training, as well as utilizing subsets that best approximate the full loss (Feldman, 2020) or gradient (Mirzasoleiman et al., 2020; Killamsetty et al., 2021). Aside from selection strategies, existing approaches vary in the ***training schemes*** which can be divided roughly into two categories: static ones and dynamic (or adaptive) ones. Static methods (Paul et al., 2021; Coleman et al., 2020; Toneva et al., 2019) decouple the subset selection and the model training, where the subset is constructed ahead and the model is trained on such a fixed subset. Dynamic methods (Mindermann et al., 2022; Killamsetty et al., 2021), however, update the subset in conjunction with the training process. Though effectively eliminates amounts of samples, it is still not well understood how these different training schemes influence the final model.

---

[*]Corresponding author
[1]We use the terms data, sample, and instance interchangeably

In this paper, we propose dynamic margin selection (DynaMS). For the selection strategy, we inquire the classification margin, namely, the distance to the decision boundary. Intuitively, samples close to the decision boundary influence more and are thus selected. Classification margin explicitly utilizes the observation that the decision boundary is mainly determined by a subset of the data. For the training scheme, we show the subset that benefits training most varies as the model evolves during training, static selection paradigm may be sub-optimal, thus dynamic selection is a better choice. Synergistically integrating classification margin selection and dynamic training, DynaMS is able to converge to the optimal solution with large probability. Moreover, DynaMS admits theoretical generalization analysis. Through the lens of generalization analysis, we show that by catching the training dynamics and progressively improving the subset selected, DynaMS enjoys better generalization compared to its static counterpart.

Though training on subsets greatly reduces the training computaiton, the overhead introduced by data evaluation undermines its significance. Previous works resort to a lighter proxy model. Utilizing a separate proxy (Coleman et al., 2020), however, is insufficient for dynamic selection, where the proxy is supposed to be able to agilely adapt to model changes. We thus propose parameter sharing proxy (PSP), where the proxy is constructed by multiplexing part of the underlying model parameters. As parameters are shared all along training, the proxy can acutely keep up with the underlying model. To train the shared network, we utilize slimmable training (Yu et al., 2019) with which a well-performing PSP and the underlying model can be obtained in just one single train. PSP is especially demanding for extremely large-scale, hard problems. For massive training data, screening informative subset with a light proxy can be much more efficient. For hard problems where model evolves rapidly, PSP timely updates the informative subset, maximally retaining the model utility.

Extensive experiments are conducted on benchmarks CIFAR-10 and ImageNet. The results show that our proposed DynaMS effectively pick informative subsets, outperforming a number of competitive baselines. Note that though primarily designed for supervised learning tasks, DynaMS is widely applicable as classifiers have become an integral part of many applications including foundation model training (Devlin et al., 2019; Brown et al., 2020; Dosovitskiy et al., 2021; Chen et al., 2020), where hundreds of millions of data are consumed.

In summary, the contributions of this paper are three-folds:

- We establish dynamic margin select (DynaMS), which selects informative subset dynamically according to the classification margin to accelerate the training process. DynaMS converges to its optimal solution with large probability and enjoys better generalization.

- We explore constructing a proxy by multiplexing the underlying model parameters. The resulting efficient PSP is able to agilely keep up with the model all along the training, thus fulfill the requirement of dynamic selection.

- Extensive experiments and ablation studies demonstrate the effectiveness of DynaMS and its superiority over a set of competitive data selection methods.

## 2 METHODOLOGY

To accelerate training, we propose dynamic margin selection (DynaMS) whose framework is presented in Figure 1. Instances closest to the classification decision boundary are selected for training, and the resulting strategy is named margin selection (MS). We show that the most informative subset changes as the learning proceeds, so that a dynamic selection scheme that progressively improves the subset can result in better generalization. Considering the computational overhead incurred by selection, we then explore parameter sharing proxy (PSP), which utilizes a much lighter proxy model to evaluate samples. PSP is able to faithfully keep up with the underlying model in the dynamics selection scheme. The notations used in this paper are summarized in Appendix H

### 2.1 SELECTION WITH CLASSIFICATION MARGIN

Given a large training set $\mathcal{T} = \{\boldsymbol{x}_i, y_i\}_{i=1}^{|\mathcal{T}|}$, data selection extracts the most informative subset $\mathcal{S} \subset \mathcal{T}$ trained on which the model $f(\boldsymbol{x})$ yields minimal performance degradation. Towards this end, we utilize the classification margin, that is, the distance to the decision boundary, to evaluate the informativeness of each sample. $|\mathcal{S}|$ examples with the smallest classification margin are selected.

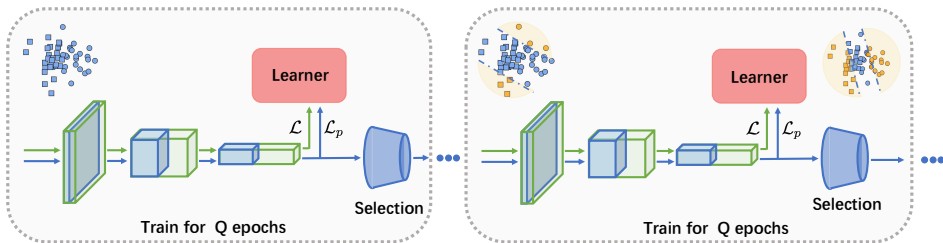

Figure 1: The overall framework of dynamic margin select with parameter sharing proxy (DynaMS+PSP). The green model indicates the underlying model to be trained and the blue one is the parameter sharing proxy which efficiently evaluates data. Instances are selected every $Q$ epochs, then the model is trained on the selected subset.

Intuitively, these samples should be influential most to the model decision. Following (Mickisch et al., 2020; Emam et al., 2021), the decision boundary between two classes $c_1$ and $c_2 \in \{1, \ldots C\}$ is $\mathcal{B} := \{x \mid f_{c_1}(x) = f_{c_2}(x)\}$, where $f_c(x)$ is the $c$ entry of model output, indicating the probability of $x$ belonging to class $c$. The classification margin is then:

$$M(x, c_1, c_2) = \min_{\delta} \|\delta\|_2 \quad \text{s.t. } x + \delta \in \mathcal{B} \tag{1}$$

which is the minimal perturbation required to move $x$ form $c_1$ to $c_2$. Directly computing the margin is infeasible for deep neural networks, so scoring is conducted in the feature space instead as in (Emam et al., 2021). Typically neural networks applies a linear classifier on top of the features (Goodfellow et al., 2016), so the classification margin $M(x, c_1, c_2)$ can be easily obtained as: $M(x, c_1, c_2) = (W_{c_1} - W_{c_2})^\top h(x) / \|W_{c_1} - W_{c_2}\|_2$, where $W \in \mathbb{R}^{d \times C}$ is the weight of the linear classifier [2] and $h(x)$ is the feature of $x$. In this way, the classification margin of a labeled sample $(x, y)$ along class $c$ is $M(x, y, c)$ if $y \neq c$ or $\min_{\tilde{c} \neq y} M(x, y, \tilde{c})$ if $y = c$. The former indicates the distance moving $(x, y)$ to class $c$ while the latter is the distance moving $(x, y)$ to the nearest class other than $y$. To keep the subset balanced, we evenly pick $|\mathcal{S}|/C$ samples with the smallest classification margin along each class. The resulting strategy is named margin selection (MS), denoted as $\mathrm{MS}(w, \mathcal{T}, |\mathcal{S}|)$. The procedure is detailed in Algorithm 1 in Appendix A.

## 2.2 DYNAMIC SELECTION

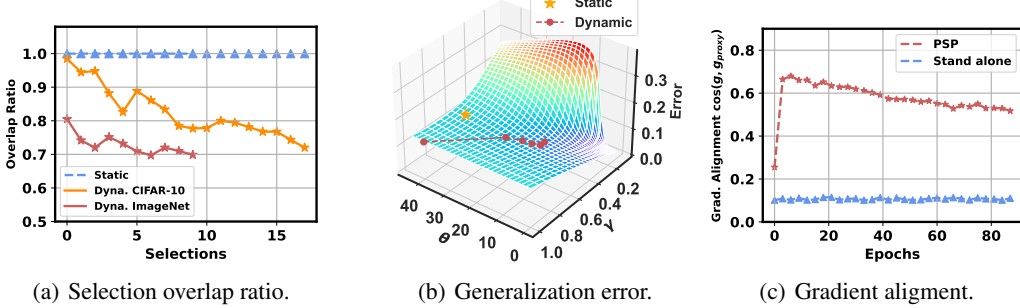

(a) Selection overlap ratio.    (b) Generalization error.    (c) Gradient aligment.

Figure 2: (a) Overlap ratio of subsets extracted in two consecutive selections. A near 1 overlap ratio means the selection is converged. (b) Generalization of static as well as dynamic data selection according to classification margin. Selection budget $\gamma_{\text{avg}} = \gamma_s = 60\%$. (c) The averaged gradient alignment $\cos(g, g_{\text{proxy}})$ of parameter sharing proxy and an stand-alone proxy along model training.

Given the subset selected, model is subsequently trained on $\mathcal{S}$. Conventional static training scheme assumes that the optimal subset converges and is not related to the model training dynamic (Paul et al., 2021; Coleman et al., 2020). Though effectively eliminate instances, the "converged optimal

---

[2]Without loss of generality, we omit the bias term for notation clarity

subset" assumption may be too strong. To investigate whether the most informative samples vary during training, we plot the overlap ratio of samples selected in two consecutive selections during the training of ResNet models, shown in Figure 2(a). We train for 200 epochs and 120 epochs on CIFAR-10 and ImageNet respectively, and conduct selection every 10 epochs. It can be observed that the overlap ratio is on average 0.83 for CIFAR-10 and 0.73 for ImageNet rather than 1.0, meaning that samples that most benefit model training vary as the model evolves. A fixed subset may be outdated after parameter updates, thus yielding sub-optimal results.

We thus resort to a dynamic scheme where data selection is performed after each $Q$ epochs training [3]. By selecting in conjunction with training, the informative subset gets updated according to the current model status. For the $k$th selection, the informative subset $\mathcal{S}_k$ is constructed by picking portion $\gamma_k$ samples so that $|\mathcal{S}_k| = \gamma_k |\mathcal{T}|$. The selection ratio $\gamma_k$ determines the critical margin $\kappa_k$, where only samples with classification margin smaller than $\kappa_k$ are kept. $\mathcal{S}_k$ will then be used for training $Q$ epochs. In the following, we provide a convergence analysis of DynaMS and show that DynaMS achieves better generalization by constantly improving the selected subset.

**Convergence Analysis**    We now study the conditions for the convergence of training loss achieved by DynaMS. We use logistic regression (LR) to demonstrate and then show the conditions are well satisfied when LR is used on top of deep feature extractors. We have the following theorem:

**Theorem.** *Consider logistic regression $f(\boldsymbol{x}) = \frac{1}{1+e^{-\boldsymbol{w}^\top \boldsymbol{x}}}$ with $N$ Gaussian training samples $\boldsymbol{x} \sim \mathcal{N}(0, \Sigma)$, $\boldsymbol{x} \in \mathbb{R}^d$. Assume $\|\boldsymbol{w}\|_2 \leq D$ and $\frac{N}{d} < \alpha$. Let $\boldsymbol{w}^*$ be the optimal parameters and $\lambda$ be the largest eigenvalue of the covariance $\Sigma$. For $t \in \{1, \dots T\}$ and constants $\varepsilon > D\sqrt{\frac{\lambda}{2}} - 1, \zeta > 1, \mu >> \alpha$, select subset with critical margin $\kappa_t = (1 + \varepsilon) \log(\zeta T - t)$ and update parameters with learning rate $\eta = \frac{DN}{E\sqrt{T}}$. Then with probability at least $1 - \frac{\alpha}{\mu}$*

$$\min_t \mathcal{L}(\boldsymbol{w}_t) - \mathcal{L}(\boldsymbol{w}^*) \leq DE\left(\frac{1}{T^{\frac{1}{4}}} + \frac{c_{\varepsilon, \zeta}}{T^{\frac{3}{4} + \varepsilon}} + \frac{c_{\varepsilon, \zeta, \lambda}}{T^{\beta}}\right) \tag{2}$$

*where $E = \sqrt{d\lambda}(1 + (2\mu)^{\frac{1}{4}})$, $\beta = \frac{(1+\varepsilon)^2}{2D^2\lambda} - \frac{1}{4}$, $c_{\varepsilon, \zeta}$ and $c_{\varepsilon, \zeta, \lambda}$ are constants depending on $\varepsilon, \zeta$ and $\lambda$.*

The proof is left in Appendix B. Theorem 2.2 indicates that dynamically selecting data based on the classification margin is able to converge and achieve the optima $\boldsymbol{w}^*$ with large probability. The Gaussian input assumption is overly strong in general, but when the linear classifier is adopted on top of a wide enough feature extractor, the condition is well satisfied because a infinitely wide neural network resembles Gaussian process (Lee et al., 2019; Xiao et al., 2018; de G. Matthews et al., 2018).

**Generalization Analysis**    Recently, (Sorscher et al., 2022) developed an analytic theory for data selection. Assume training data $\boldsymbol{x}_i \sim \mathcal{N}(0, \mathbf{I})$ and there exists an oracle model $\boldsymbol{w}_o \in \mathbb{R}^d$ which generates the labels such that $y_i = \text{sign}\left(\boldsymbol{w}_o^\top \cdot \boldsymbol{x}_i\right)$. Following static selection, when an estimator $\boldsymbol{w}$ is used to pick samples that have a small classification margin, the generalization error takes the form $\mathcal{E}(\alpha, \gamma, \theta)$ in the high dimensional limit. $\alpha = \frac{|\mathcal{T}|}{d}$ indicates the abundance of training samples before selection; $\gamma$ determines the selection budget and $\theta = \arccos\left(\frac{\boldsymbol{w}^\top \boldsymbol{w}_o}{\|\boldsymbol{w}\|_2 \cdot \|\boldsymbol{w}_o\|}\right)$ shows the closeness of the estimator to the oracle. The full set of self-consistent equations characterizing $\mathcal{E}(\alpha, \gamma, \theta)$ is given in Appendix C. By solving these equations the generalization error $\mathcal{E}(\alpha, \gamma, \theta)$ can be obtained.

We then extend it to the dynamic scheme. For the $k$th selection, we use the model trained on $\mathcal{S}_{k-1}$ as the estimator $\boldsymbol{w}_k$, which deviates from oracle by angle $\theta_k = \arccos\left(\frac{\boldsymbol{w}_{k-1}^\top \boldsymbol{w}_o}{\|\boldsymbol{w}_{k-1}\|_2 \cdot \|\boldsymbol{w}_o\|_2}\right)$, to evaluate and select samples. The resulting subset $\mathcal{S}_k$ will be used for subsequent training of model $\boldsymbol{w}_{k+1}$, which will later be used as an estimator at $k + 1$ to produce $\mathcal{S}_{k+1}$. In this way, generalization of dynamic scheme can be obtained by recurrently solving the equations characterizing $\mathcal{E}(\alpha, \gamma_k, \theta_k)$ with updated keeping ratio $\gamma_k$ and estimator deviation $\theta_k$. Note that in each round of selection, samples are picked with replacement, so the abundance of training samples $\alpha$ is kept fixed. The keeping ratio $\gamma_k$, determining the subset size, can be scheduled freely to meet various requirements.

---

[3]For extremely large dataset case where training can be accomplished within just one or a few epochs, the selection can be performed every $Q$ iterations

We compare the generalization of dynamic selection and its static counterpart in Figure 2(b). We show the landscape of $\mathcal{E}(\alpha, \gamma, \theta)$ with different $\gamma$ and $\theta$ by solving the generalization equations numerically. $\alpha = 3.2$ is kept fixed, which means the initial training data is abundant; We use static training with $\theta_s = 40°$ and $\gamma_s = 0.6$ as control group. To make the comparison fair, we make sure $\frac{1}{K}\sum_{k=1}^{K} |\gamma_k| = \gamma_s$, so that the averaged number of samples used in the dynamic scheme equals the subset size used in the static scheme. From Figure 2(b), we see that in dynamic selection, the estimator gets constantly improved ($\theta_k$ decreases), so that the subsets get refined and the model achieves better generalization. Discussion on selecting with different $\alpha, \gamma$ and $\theta$ is given in Appendix D.

### 2.3 PARAMETER SHARING PROXY

With dynamic selection, the number of updates is reduced. However, the computational overhead incurred by data selection undermines its significance, especially when the model is complex and samples are evaluated frequently. Aside from designing efficient selection strategies, previous works explored utilizing a lighter model as proxy to evaluate the instances so that the problem can be ameliorated. Pretrain a separate proxy and evaluate instances prior to model training (Coleman et al., 2020), however, is insufficient for dynamic selection, as a static proxy can not catch the dynamics of the underlying model. A proxy that fulfills the requirements of dynamic selection is still absent.

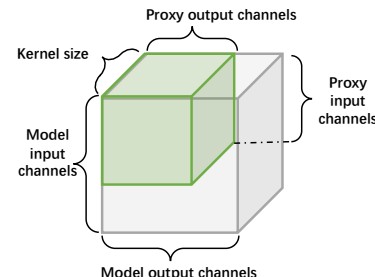

Figure 3: Parameter sharing proxy which is constructed with part of model parameters.

We thus propose parameter sharing proxy (PSP), where part of the model is used as the proxy. Taking convolutional neural network as an example, for a layer with kernel $\mathcal{W} \in \mathbb{R}^{c_i \times c_o \times u \times u}$, where $c_i$, $c_o$ and $u$ are number of input filters, number of output filters and kernel size respectively, the corresponding kernel of proxy is then: $\mathcal{W}_{\text{proxy}} = \mathcal{W}_{1:pc_i, 1:pc_o, :, :}$, where $p \in [0, 1]$ is a slimming factor. As shown in Figure 3, the proxy kernel is constructed with the first $pc_i$ input channels and first $pc_o$ output channels. A $p$ times thinner proxy can be obtained by applying $p$ to each layer.

With separate batch normalization for proxy and model, PSP forms a slimmable network (Yu et al., 2019), where multiple models of different widths are jointly trained and they all yield good performance. As the parameters are shared, the proxy can acutely keep up with the model change, thus applicable for dynamic selection. We further investigate the gradients alignment of the proxy and the original model through their cosine similarity:

$$\cos(\boldsymbol{g}, \boldsymbol{g}_{\text{proxy}}) = \frac{\boldsymbol{g}^\top \boldsymbol{g}_{\text{proxy}}}{\|\boldsymbol{g}\|_2 \cdot \|\boldsymbol{g}\|_2}, \text{ where } \boldsymbol{g} = \nabla_{\mathcal{W}}\mathcal{L}\left(\mathcal{W}\right), \boldsymbol{g}_{\text{proxy}} = \nabla_{\mathcal{W}}\mathcal{L}\left(\mathcal{W}_{\text{proxy}}\right) \quad (3)$$

A positive cosine value indicates $\boldsymbol{g}_{\text{proxy}}$ stands in the same side with $\boldsymbol{g}$, thus updates on proxy and the model benefits each other. We compare the gradient alignment of PSP and a stand-alone proxy in Figure 2(c) on ResNet-50. With $p = 0.5$, we see that $\cos(\boldsymbol{g}, \boldsymbol{g}_{\text{proxy}})$ for PSP is much larger than the stand-alone proxy. Given the well-aligned gradients, PSP requires fewer training epochs. Overall workflows of DynaMS and DynaMS+PSP is shown in Algorithm 2 and Algorithm 3 of Appendix A. PSP is especially advantageous for large and hard problems. When the data is extremely large, training PSP on a small subset is cheaper than evaluating the extremely large training set with the original model, making it much more efficient. When the task is hard and model changes rapidly during training, PSP can timely updates the informative subset, maximally retaining the model utility.

## 3 RELATED WORK

Accelerating training by eliminating redundant training instances has long been a research focus in academia. This is accomplished by adopting an effective selection strategy and an appropriate training scheme. We summarize the related literature from these two strands of research in the following.

**Selection Strategy** Sample selection can be accomplished with various principles. (Loshchilov & Hutter, 2015; Jiang et al., 2019; Paul et al., 2021) tend to pick samples that incur large loss or gradient norm (CE-loss, EL2N, GraNd). (Toneva et al., 2019) inspects the "unforgettable"

examples that are rarely misclassified once learned, and believes these samples can be omitted without much performance degradation. Other works adopt uncertainty. Samples with the least prediction confidence are preferred (Settles, 2010). Recently, (Mirzasoleiman et al., 2020; Killamsetty et al., 2021) select subset that best covers or approximates the full gradient (Craig, GradMatch). However, these requires per-sample gradient as well as an additional optimization which is expensive both in run-time and in memory. Our work utilizes the classification margin to identify informative samples,

Table 1: Computational and space complexity of different selection strategies. For GraNd, Craig and GradMatch, only gradients of the classification layer are considered to avoid overly large complexity.

| Strategy | Time Complex. | Space Complex. |
|---|---|---|
| CE-loss | $\mathcal{O}((C \cdot d + \log|\mathcal{S}|) \cdot |\mathcal{T}|)$ | $\mathcal{O}(|\mathcal{T}|)$ |
| EL2N | $\mathcal{O}((C \cdot d + \log|\mathcal{S}|) \cdot |\mathcal{T}|)$ | $\mathcal{O}(|\mathcal{T}|)$ |
| GraNd | $\mathcal{O}((C \cdot d + \log|\mathcal{S}|) \cdot |\mathcal{T}|)$ | $\mathcal{O}(|\mathcal{T}|)$ |
| Craig | $\mathcal{O}(C \cdot d \cdot |\mathcal{T}| \cdot |\mathcal{S}|)$ | $\mathcal{O}(|\mathcal{S}| \cdot |\mathcal{T}|)$ |
| GradMatch | $\mathcal{O}(C \cdot d \cdot |\mathcal{T}| \cdot |\mathcal{S}|)$ | $\mathcal{O}(C \cdot d \cdot |\mathcal{T}|)$ |
| MS | $\mathcal{O}(C \cdot (d + \log|\mathcal{S}|) \cdot |\mathcal{T}|)$ | $\mathcal{O}(C \cdot |\mathcal{T}|)$ |

which is efficient and can synergistically adapt to various training schemes. Comparison of these strategies is given in Table 1, where $d$ is the dimension of data feature. MS is slightly slower than selection via loss (CE-loss and EL2N), but much more efficient than Craig and GradMatch. Here we consider only the complexity of the selection strategy itself, time spent for feature extraction is not included. Classification margin has been previously explored in the active learning literature (Ducoffe & Precioso, 2018; Emam et al., 2021), here we utilize it for training acceleration.

**Training Schemes**   Data selection brings more options to training. Under the conventional static training scheme (Paul et al., 2021; Toneva et al., 2019; Coleman et al., 2020), data selection is conducted prior to model update, and the informative subset is kept fixed. Contrastively, online batch selection picks batch data each iteration (Loshchilov & Hutter, 2015; Alain et al., 2015; Zhang et al., 2019; Mindermann et al., 2022). Though sufficiently considered the training dynamics, the overly frequent sample evaluation incurs prohibitive computational overhead. Recently, (Killamsetty et al., 2021) tried selecting after several epochs' training, which is similar to our dynamic scheme. However, the dynamic training scheme is just utilized as a compromise to avoid overly frequent selection. A formal analysis of its advantage over the static scheme is absent.

By systematically considering the selection strategy, the model training, as well as the proxy design, Our proposed DynaMS forms an effective data selection framework for efficient training.

## 4 EXPERIMENTS

In this section, we first analyse the effectiveness of each design ingredient in Section 4.2. Then we compare to state-of-the-art algorithms in Section 4.3. Code is available at `https://github.com/ylfzr/DynaMS-subset-selection`.

### 4.1 EXPERIMENTAL SETUP

We conduct experiments on CIFAR-10 Krizhevsky & Hinton (2009) and ImageNet Jia et al. (2009), following standard data pre-processing in He et al. (2016). A brief summarization of the experimental setup is introduced below, while complete hyper-parameter settings and implementation details can be found in Appendix F.

**CIFAR-10 Experiments**   For CIFAR-10, we train ResNet-18 (He et al., 2016) for 200 epochs. Selection is conducted every 10 epochs, so overall there are 19 selections ($K = 19$). For subset size, we adopt a simple linear schedule: $\gamma_k = 1 - k \cdot a$ for $k = 1, \ldots, K$, where $a$ determines the reduction ratio. We make sure $\gamma_{\text{avg}} = \frac{1}{K}\sum_{k=1}^{K} \gamma_k = \gamma_s$. In this way, the averaged number of data used in the dynamic scheme ($\gamma_{\text{avg}}$) is kept equal to that of static training ($\gamma_s$) for fair comparison. For $0.6\times$ acceleration, $a = 0.042$. We conduct experiments on a NVIDIA Ampere A-100.

**ImageNet Experiments**   For ImageNet, we choose ResNet-18 and ResNet-50 as base models. Following the conventions, the total training epoch is 120. Selection is also conducted every 10 epochs, so altogether $K = 11$. For subset size, aside from the linear schedule, we also explore a power schedule where $\gamma_k$ decays following a power law: $\gamma_k = m \cdot k^{-r} + b$ for $k = 1, 2, \ldots, K$. For $0.6\times$ acceleration, we set $m = 0.398$, $r = 0.237$ and $b = 0.290$. Please see Appendix F for

more details. The power schedule reserves more samples in late training, preventing performance degradation caused by over data pruning. We conduct experiments on four NVIDIA Ampere A-100s.

## 4.2 ABLATION STUDIES

Table 2: Comparison of different data selection strategies. Except for DynaMS, all the other methods conduct training in the conventional static scheme.

| Strategy | Top1 Acc. | Top5 Acc. |
|----------|-----------|-----------|
| CE-loss  | 72.073    | 91.727    |
| EL2N     | 72.032    | 91.778    |
| MS       | 72.888    | 91.807    |
| DynaMS   | **74.558**| **92.334**|

Table 3: Accuracy of utilizing parameter sharing proxy (PSP) with different width configurations. FLOPs↓ is the reduction of FLOPs required when use proxy for sample evaluation.

| Width  | Top1 Acc. | Proxy Top1 Acc. | FLOPs ↓ |
|--------|-----------|-----------------|---------|
| 1.00×  | 74.558    | -               | -       |
| 0.75×  | 73.694    | 72.882          | 43.75%  |
| 0.50×  | 73.401    | 70.720          | 75.00%  |
| 0.25×  | 73.390    | 62.349          | 93.75%  |

We use ResNet-50 on ImageNet to illustrate the effect of each ingredient in DynaMS, that is, the classification margin criteria, the dynamic training scheme as well as the parameter sharing proxy.

**The effect of classification margin selection**   To inspect the effect of classification margin selection (MS), we compare MS against two widely applied selection strategies **CE-loss** (Loshchilov & Hutter, 2015; Jiang et al., 2019) and **EL2N** (Paul et al., 2021). CE-loss selects samples explicitly through the cross-entropy loss they incur while EL2N picks samples that incur large L2 error. We compare the three under the conventional static scheme so any other factors aside from the selection strategy is excluded. Samples are evaluated after 20 epochs of pretraining. The model is then reinitialized and trained on the selected subset, which contains 60% original samples. As shown in Table 2, MS achieves the best accuracy among the three, validating its effectiveness.

**The effect of dynamic training**   We then apply dynamic selection on MS, where the average subset size is also kept to be 60% of the original dataset. From Table 2 we see that DynaMS outperforms MS by 1.67%, which is significant on large scale dataset like ImageNet. The superiority of DynaMS validates that by constantly improving the model and updating the subset, dynamic selection scheme can result in better performance. Note that DynaMS can be more practical since it does not require the 20 epochs training prior to selection as required in the static scheme.

**The effect of parameter sharing proxy**   We now study the parameter sharing proxy (PSP). An effective proxy is supposed to be faithful, and can agilely adapt to model updates. In Figure 4, we plot the Spearman rank correlation as well as the overlap ratio of samples selected with the proxy and the model. We see that all along the training, the rank correlation is around 0.68, and over 78% samples selected are the same, indicating that the proxy and the model are fairly consistent. We then investigate how will the complexity, measured by floating point operations (FLOPs), of proxy affect. We enumerate over the slimming factor $p \in \{0.25, 0.5, 0.75, 1.0\}$ to construct proxies of different widths, the corresponding FLOPs are 6.25%, 25.00%, 56.25%, 100% respectively. In Table 3, we see that significant computation reduction can be achieved with moderate performance degradation.

## 4.3 COMPARISONS WITH STATE-OF-THE-ARTS

Finally, we compare DynaMS against various state-of-the-art methods. Aside from **CE-loss** and **EL2N**, **Random** picks samples uniformly at random. **GraNd** (Paul et al., 2021) select samples that incur large gradient norm. **Forget** (Toneva et al., 2019) counts how many times a sample is mis-classified (forget) after it is learned. Samples more frequently forgotten are preferred. We evaluate the forget score after 60 epochs training. To avoid noisy evaluation, many of these static selection approaches ensembles networks before selection. The number of ensambled models is given by the subscription. **Auto-assist** (Zhang et al., 2019) select samples that incur large loss value on a small proxy. Selection is conducted in each iteration thus forming an online batch selection (OLBS) scheme. **DynaCE** and **DynaRandom** apply the corresponding selection strategy, but are trained in a dynamic way. **CRAIG** and **GradMatch** propose to reweight and select subsets so that they best cover or approximate the full gradient. In the experiments, we use the per-batch variant of CRAIG and

Table 4: Comparison for ResNet-18 on Cifar-10.

| Methods | Types | Budget | Schedule | Acc. | Time(min) |
|---|---|---|---|---|---|
| Original | - | 100% | - | 95.52 | 28.6 |
| Random | Stat. | 60% | - | 94.09 | 17.4 |
| $EL2N_1$ | Stat. | 60% | - | 94.55 | $17.4_{+2.9}$ |
| $EL2N_{10}$ | Stat. | 60% | - | 95.34 | $17.4_{+29.5}$ |
| $GraNd_{10}$ | Stat. | 60% | - | 95.21 | $17.4_{+29.5}$ |
| $Forget_{10}$ | Stat. | 60% | - | 95.29 | $17.4_{+88.5}$ |
| OnlineMS | OLBS. | 60% | Const. | 95.21 | 9076.3 |
| Auto-assist | OLBS. | 60% | Const. | 92.37 | 24.6 |
| DynaRandom | Dyna. | 60% | Linear | 94.45 | 20.9 |
| DynaCE | Dyna. | 60% | Linear | 94.96 | 21.0 |
| Craig | Dyna. | 60% | Const. | 94.36 | 28.7 |
| GradMatch | Dyna. | 60% | Const. | 94.84 | 26.7 |
| DynaMS | Dyna. | 60% | Linear | 95.28 | 21.3 |

Figure 4: Correlation of proxy and model.

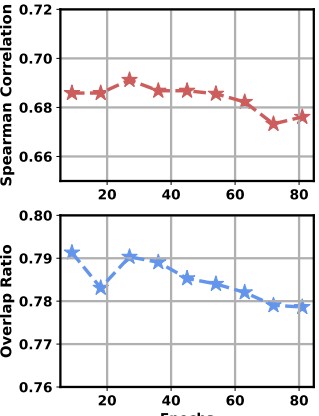

GradMatch proposed in (Killamsetty et al., 2021) with 10 epoch warm start [4]. The two approaches utilize dynamic selection scheme, all the training settings are kept the same as our DynaMS.

In table 4, average accuracy from 5 runs on CIFAR-10 as well as their running time are reported. Due to limited space, the standard deviation is given in Appendix E. We see that DynaMS achieves comparable performance against the strongest baselines ($EL2N_{10}$, $GraNd_{10}$, $Forget_{10}$) while being more efficient. Note that the static methods require pretraining one or several models for 20 epochs before selection. Considering this cost (subscript of the reported running time), the acceleration of these methods is less significant. We also compare two online batch selection methods, OnlineMS and Auto-assist (Zhang et al., 2019). OnlineMS picks samples with MS, but the selection is conducted each iteration. OnlineMS didn't outperform DynaMS, meaning more frequent selection is not necessary. Rather, selecting at each optimization step incurs prohibitive computational overhead. Auto-assist didn't get good performance in this experiment. This may results from the overly simple proxy. The logistic regression proxy adopted may not sufficiently evaluate the candidate samples.

Table 5: Comparison for ResNet-18 and ResNet-50 on ImageNet.

| | Types | Budget | Schedule | ResNet 18 | | | ResNet 50 | | |
|---|---|---|---|---|---|---|---|---|---|
| | | | | Acc@1 | Acc@5 | Time(hrs) | Acc@1 | Acc@5 | Time(hrs) |
| Original | - | 100% | - | 70.56 | 89.95 | 12.8 | 75.96 | 92.75 | 18.2 |
| Random | Stat. | 60% | - | 67.16 | 87.50 | 7.6 | 72.4 | 90.85 | 10.9 |
| $EL2N_1$ | Stat. | 60% | - | 66.38 | 88.56 | $7.6_{+2.1}$ | 72.03 | 91.78 | $10.9_{+3.0}$ |
| $EL2N_{10}$ | Stat. | 60% | - | 66.46 | 88.73 | $7.6_{+21.1}$ | 72.18 | 92.02 | $10.9_{+29.7}$ |
| $GraNd_{10}$ | Stat. | 60% | - | 66.50 | 88.76 | $7.6_{+21.1}$ | 72.14 | 92.16 | $10.9_{+29.7}$ |
| $Forget_{10}$ | Stat. | 60% | - | 67.84 | 87.50 | $7.6_{+63.1}$ | 73.50 | 91.41 | $10.9_{+89.1}$ |
| SVP+Forget | Stat. | 60% | - | - | - | - | 72.90 | 91.37 | $10.9_{+12.8}$ |
| SVP+Entropy | Stat. | 60% | - | - | - | - | 73.00 | 91.52 | $10.9_{+12.8}$ |
| DynaRandom | Dyna. | 60% | Power | 67.59 | 87.62 | 8.9 | 72.63 | 90.91 | 12.1 |
| DynaCE | Dyna. | 60% | Power | 67.58 | 88.10 | 9.2 | 72.80 | 91.31 | 12.5 |
| Craig | Dyna. | 60% | Const. | 65.32 | 86.92 | 12.1 | 70.69 | 90.72 | 16.0 |
| GradMatch | Dyna. | 60% | Const. | 66.48 | 88.61 | 11.7 | 71.79 | 91.67 | 15.2 |
| DynaMS | Dyna. | 60% | Linear | 68.12 | 88.93 | 9.6 | 74.10 | 92.25 | 12.9 |
| DynaMS | Dyna. | 60% | Power | **68.65** | **89.21** | 9.6 | **74.56** | **92.33** | 13.0 |
| DynaMS+PSP | Dyna. | 60% | Linear | - | - | - | 73.59 | 91.79 | 12.8 |
| DynaMS+PSP | Dyna. | 60% | Power | - | - | - | 73.40 | 91.80 | 12.8 |

[4]For cifar-10, we use the published implementation from https: //github.com/decile-team/cords. For ImageNet, we modify the implementation to the distributed setting.

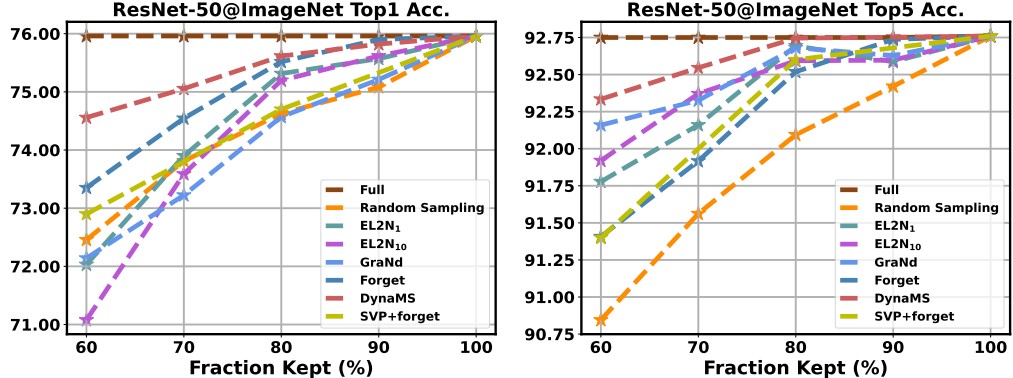

Figure 5: Comparison under different (on average) sample budgets.

For ImageNet, we also report the average accuracy from 5 runs as well as their running time. The standard deviation is given in Appendix E DynaMS outperforms all the baselines. For instance, it achieves 68.65% and 74.56% top-1 accuracy given on average 60% samples for ResNet-18 and ResNet-50 respectively, surpassing the most competitive counterpart Forget by 0.81% and 1.06%. Compared to the static methods which require additional pretraining, 60 epochs for Forget and 20 for the others, DynaMS is much more efficient. CRAIG and GradMatch didn't get good performance on ImageNet. This might because we use the per-batch variant in (Killamsetty et al., 2021), and set batch size 512 in order to fit the per-sample gradients into memory. The per-batch variant treats each mini-batch as one sample and selects mini-batches during the gradient matching process. So a larger batch size means more coarse grain selection which may lead to inferior performance. We also compare a variant DynaRandom. DynaRandom adopts the dynamic selection scheme but a random subset is constructed at each selection. DynaMS outperforms DynaRandom by 1.06% and 1.93% for ResNet-18 and ResNet-50 respectively, indicating that the superiority of DynaMS over static methods comes from effectively identifying informative samples instead of witnessing more data.

ResNet-50 is rather complex and the data evaluation time is non-negligible. We thus apply parameter sharing proxy to reduce the evaluation time. The proxy is $0.5\times$ width so the evaluation requires around $0.25\times$ computation compared to the original model. As the gradients of the proxy and the underlying model are well aligned, we only train DynaMS+PSP for 90 epochs. From table 5, though utilizing a proxy harms performance compared to DynaMS, it still outperforms all the other baselines. Specifically, **SVP** also uses a proxy for sample evaluation. The proxy, however, is a statically fully trained ResNet-18. The superiority of DynaMS+PSP over SVP shows the necessity of a dynamic proxy that agilely keeps up with the change of underlying model. The advantages of DynaMS+PSP over DynaMS on efficiency can be significant for extremely large scale problems where massive data is available while only a small fraction of data is sufficient for training. To further demonstrate DynaMS, we draw the accuracy curvature of ResNet-50 against different (on average) sample budgets from 60% to 100% in Figure 5. It can be found that our DynaMS consistently outperforms all the other data selection strategies on different budgets. Finally, To get a better understanding of how the selected samples look like and how they change over time, we visualize samples picked in different selection steps along the training. See G for more details.

## 5 CONCLUSION

In this paper, we propose DynaMS, a general dynamic data selection framework for efficient deep neural network training. DynaMS prefers samples that are close to the classification boundary and the selected "informative" subset is dynamically updated during the model training. DynaMS has a high probability to converge and we pioneer to show both in practice and theory that dynamic selection improves the generalization over previous approaches. Considering the additional computation incurred by selection, we further design a proxy available for dynamic selection. Extensive experiments and analysis are conducted to demonstrate the effectiveness of our strategy.

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

# A APPENDIX

## A ALGORITHM PROCEDURE

Algorithm 1 outlines the procedure of margin selection (MS). In MS, distances of the current sample $(\boldsymbol{x}, y)$ to each other class $c$ are computed. If $y \neq c$, the classification margin of $(\boldsymbol{x}, y)$ and class $c$ is $M(\boldsymbol{x}, y, c)$, which is the distance of moving $\boldsymbol{x}$ from class $y$ to class $c$. If $y = c$, the classification margin is $\min_{\tilde{c} \neq y} M(\boldsymbol{x}, y, \tilde{c})$, which corresponds to the distance moving $(\boldsymbol{x}, y)$ to another class that is the most close to $\boldsymbol{x}$. For the whole candidate set $\mathcal{T}$, this generates a $|\mathcal{T}| \times C$ score matrix. After the classification margins are obtained, $|\mathcal{S}|/C$ samples with the smallest classification margin along each class are picked. This keeps samples collected in the subset balanced.

---

**Algorithm 1** Margin selection: $\mathrm{MS}(\mathrm{w}, \mathcal{T}, \gamma)$

---

**Input:**
    Candidate set $\mathcal{T}$, keeping ratio $\gamma$, number of classes $C$;
    Network with weights $\boldsymbol{w}$, including weights of the final classification layer $\boldsymbol{W}$;
**Output:**
    Selected subset according to the classification margin $\mathcal{S}$.
1: Compute the keeping budget $|\mathcal{S}| = \gamma \cdot |\mathcal{T}|$, initialize the subset $\mathcal{S} = \{\}$

    *// Evaluating: compute the classification margin.*
2: **for** $(\boldsymbol{x}, y) \in \mathcal{T}$ **do**
3:     **for** $c = 1 : C$ **do**
4:         Compute the classification margin of the sample to the $(\boldsymbol{y}, c)$ boundary:

$$M(\boldsymbol{x}, y, c) = \begin{cases} \min_{\tilde{c} \neq y} M(\boldsymbol{x}, y, \tilde{c}) & y = c \\ M(\boldsymbol{x}, y, c) & y \neq c \end{cases} \qquad (4)$$

5:     **end for**
6: **end for**

    *// Selecting: pick the samples according to classification margin (Equation 4.)*
7: **for** $c = 1 : C$ **do**
8:     Pick $|\mathcal{S}|/C$ samples which have the smallest classification margins ($M(\cdot)$): $\mathrm{Top}_{|\mathcal{S}|/C}(c)$.
9:     $\mathcal{S} = \mathcal{S} \bigcup \mathrm{Top}_{|\mathcal{S}|/C}(c)$
10:     Remove the already selected samples from the candidate set: $\mathcal{T} = \mathcal{T} - \mathrm{Top}_{|\mathcal{S}|/C}(c)$
11: **end for**

---

**Algorithm 2** Dynamic margin selection (DynaMS)

---

**Input:**
    Training data $\mathcal{T}$;
    Base network with weights $\boldsymbol{\mathcal{W}}$, learning rate $\eta$
    Keep ratio of each selection $\gamma_k$ where $k = 1, ..., K$, selection interval Q
**Output:**
    Model efficiently trained on selected subsets.

1: $k = 1; \gamma_k = 1$ thus $\mathcal{S}_k = \mathcal{T}$
2: **for** epochs $t = 1, ..., T$ **do**
3:     **if** $t \% Q == 0$ **then**
4:         Select subset, $\mathcal{S}_k = \mathrm{MS}(\boldsymbol{\mathcal{W}}^t, \mathcal{T}, \gamma_k)$.
5:         $k = k + 1$
6:     **else**
7:         Keep subset $\mathcal{S}_k$.
8:     **end if**
9:     Update $\boldsymbol{\mathcal{W}}$ via stochastic gradient descent on $\mathcal{S}_k$.
10: **end for**

---

---

**Algorithm 3** Dynamic margin selection (DynaMS) with parameter sharing proxy (PSP)

---

**Input:**
    Training data $\mathcal{T}$;
    Base network with weights $\mathcal{W}$, learning rate $\eta$
    Keep ratio of each selection $\gamma_k$ where $k = 1, ..., K$, selection interval Q
    Slimming factor of the proxy $r$, thus the proxy weights $\mathcal{W}_{\text{proxy}}$ is determined.
**Output:**
    Model efficiently trained on selected subsets.

1:  $k = 1; \gamma_k = 1$ thus $\mathcal{S}_k = \mathcal{T}$
2:  **for** epochs $t = 1, ..., T$ **do**
3:    **if** $t \% Q == 0$ **then**
4:      Select subset, $\mathcal{S}_k = \text{MS}(\mathcal{W}_{\text{proxy}}^t, \mathcal{T}, \gamma_k)$.
5:      $k = k + 1$
6:    **else**
7:      Keep subset $\mathcal{S}_k$.
8:    **end if**
9:    Update $\mathcal{W}$ via optimizing $\mathcal{L}(\mathcal{W}) + \mathcal{L}(\mathcal{W}_{\text{proxy}})$ on $\mathcal{S}_k$. (Slimmable training)
10: **end for**

---

A full workflow of efficient training with the proposed dynamic margin selection (DynaMS) is shown in Algorithm 2. The model is first trained on the full dataset $\mathcal{T}$ for $Q$ epochs to warm up. Subset selection kicks in each $Q$ epochs, samples are evaluated with the current model so the informative subset gets updated according to the distance of samples to the classification boundary. After selection, the model is trained on the selected subset until the next selection. The workflow incorporating parameter sharing proxy is shown in Algorithm 3. Different from naive DynaMS, samples are evaluated and selected with the proxy instead of the underlying model. During the $Q$ epochs' training, the proxy and the original model are updated simultaneously with slimmable training (Yu et al., 2019).

## B   Proof for Theorem 2.2

To prove Theorem 2.2, we first inspect the norm of $\boldsymbol{x}$. We get the following lemma.

**Lemma 1.** *For Gaussian data $\boldsymbol{x} \sim \mathcal{N}(0, \Sigma)$, let $\mu > 0$, $T > 1$ be constants, $d$ the dimension of $\boldsymbol{x}$ and $\lambda$ the largest eigenvalue of the covariance $\Sigma$, then with probability at least $1 - \frac{1}{\mu T d}$,*
$$\|\boldsymbol{x}\|_2 < \sqrt{d\lambda}(1 + (2\mu)^{\frac{1}{4}})T^{\frac{1}{4}}.$$

*Proof of Lemma 1.* For $\boldsymbol{x} \sim \mathcal{N}(0, \Sigma)$, $\|\boldsymbol{x}\|_2^2$ follows a generalized chi-squared distribution. The mean and variance can be computed explicitly as $\mathbb{E}[\|\boldsymbol{x}\|_2^2] = \text{tr}\Sigma = \sum_j \lambda_j$ and $\text{Var}(\|\boldsymbol{x}\|_2^2) = 2\text{tr}\Sigma^2 = 2\sum_j \lambda_j^2$. By Chebyshev's inequality, we have

$$Pr\left(\|\boldsymbol{x}\|_2^2 < \sum \lambda_j + \sqrt{\mu T d}\sqrt{2\sum_j \lambda_j^2}\right) > 1 - \frac{1}{\mu T d}$$

where $\mu > 0$ and $T > 1$ are constants and $d$ is the dimension of $x$. Then

as $\sum \lambda_j + \sqrt{\mu T d}\sqrt{2\sum_j \lambda_j^2} \leq (1 + \sqrt{2\mu T})d\lambda$ where $\lambda = \max_j \lambda_j$ is the largest eigenvalue of the covariance $\Sigma$, we have:

$$Pr\left(\|\boldsymbol{x}\|_2 < \sqrt{d\lambda}(1 + (2\mu)^{\frac{1}{4}})T^{\frac{1}{4}}\right) > 1 - \frac{1}{\mu T d} \tag{5}$$

$\square$

Then we can start proving Theorem 2.2.

**Theorem.** *Consider logistic regression $f(\boldsymbol{x}) = \frac{1}{1+e^{-\boldsymbol{w}^\top \boldsymbol{x}}}$ with $N$ Gaussian training samples $\boldsymbol{x} \sim \mathcal{N}(0, \Sigma)$, $\boldsymbol{x} \in \mathbb{R}^d$. Assume $\|\boldsymbol{w}\|_2 \leq D$ and $\frac{N}{d} < \alpha$. Let $\boldsymbol{w}^*$ be the optimal parameters and $\lambda$ be*

*the largest eigenvalue of the covariance* $\Sigma$. *For* $t \in \{1, \ldots T\}$ *and constants* $\varepsilon > D\sqrt{\frac{\lambda}{2}} - 1, \zeta > 1, \mu >> \alpha$, *select subset with critical margin* $\kappa_t = (1 + \varepsilon)\log(\zeta T - t)$ *and update parameters with learning rate* $\eta = \frac{DN}{E\sqrt{T}}$. *Then with probability at least* $1 - \frac{\alpha}{\mu}$

$$\min_t \mathcal{L}(\boldsymbol{w}_t) - \mathcal{L}(\boldsymbol{w}^*) \leq DE\left(\frac{1}{T^{\frac{1}{4}}} + \frac{c_{\varepsilon,\zeta}}{T^{\frac{3}{4}+\varepsilon}} + \frac{c_{\varepsilon,\zeta,\lambda}}{T^\beta}\right) \tag{6}$$

*where* $E = \sqrt{d\lambda}(1 + (2\mu)^{\frac{1}{4}})$, $\beta = \frac{(1+\varepsilon)^2}{2D^2\lambda} - \frac{1}{4}$, $c_{\varepsilon,\zeta}$ *and* $c_{\varepsilon,\zeta,\lambda}$ *are constants depending on* $\varepsilon, \zeta$ *and* $\lambda$.

*Proof of Theorem 2.2.* For logistic regression $f(\boldsymbol{x}) = \frac{1}{1+e^{-\boldsymbol{w}^\top \boldsymbol{x}}}$ with loss function

$$\mathcal{L} = \frac{1}{N}\sum_{i=1}^N \ell_i = \frac{1}{N}\sum_{i=1}^N -y_i \log \hat{y}_i - (1 - y_i)\log(1 - \hat{y}_i) \tag{7}$$

Where $\hat{y}_i$ is the predicted value. The gradient incurred training on the selected subset is then:

$$\frac{\partial \mathcal{L}_\kappa}{\partial \boldsymbol{w}} = \frac{1}{N}\sum_{i=1}^N (\hat{y}_i - y_i)\boldsymbol{x}_i \cdot \mathbf{I}(|\boldsymbol{w}^\top \boldsymbol{x}_i| < \kappa)$$

For those $|\boldsymbol{w}^\top x_i| \geq \kappa$ or "easy" samples, we have $|\operatorname{sgn}(y_i - \frac{1}{2}) \cdot \boldsymbol{w}^\top \boldsymbol{x}_i| \geq \kappa$ and with probability at least $1 - \frac{1}{\mu T d}$

$$\left\|\frac{\partial \ell_i}{\partial \boldsymbol{w}}\right\|_2 \leq \begin{cases} \frac{E \cdot T^{\frac{1}{4}}}{1+e^\kappa} & \text{if } \operatorname{sgn}(y_i - \frac{1}{2}) \cdot \boldsymbol{w}^\top \boldsymbol{x}_i \geq \kappa \\ E \cdot T^{\frac{1}{4}} & \text{if } \operatorname{sgn}(y_i - \frac{1}{2}) \cdot \boldsymbol{w}^\top \boldsymbol{x}_i \leq -\kappa \end{cases} \tag{8}$$

where $E = \sqrt{d\lambda}(1 + (2\mu)^{\frac{1}{4}})$. Note that the condition $\operatorname{sgn}(y_i - \frac{1}{2}) \cdot \boldsymbol{w}^\top x_i \leq -\kappa$ means $x_i$ is misclassified by $\boldsymbol{w}$ as well as the margin is at least $\kappa$. Denote the portion of this kind of misclassified sample in the whole training set by $r$, we have the estimate of the gradient gap

$$\operatorname{Err}_t = \left\|\frac{\partial \mathcal{L}_\kappa}{\partial \boldsymbol{w}} - \frac{\partial \mathcal{L}}{\partial \boldsymbol{w}}\right\|_2 = \frac{1}{N}\left\|\sum_{|\boldsymbol{w}^\top \boldsymbol{x}| \geq \kappa} \frac{\partial \ell}{\partial \boldsymbol{w}}(\boldsymbol{x})\right\|_2 \tag{9}$$

$$\leq \frac{ET^{\frac{1}{4}}(1 - \gamma_t)}{1 + e^{\kappa_t}} + ET^{\frac{1}{4}}(1 - \gamma_t)r_t$$

Where $\gamma_t$ is the fraction of data kept by selecting with margin $\kappa_t$. The inequality holds with probability at least $(1 - \frac{1}{\mu T d})^N > 1 - \frac{\alpha}{\mu T}$ because of Equation 8.

Note that Lemma 1 also suggest $\left\|\frac{\partial \ell}{\partial \boldsymbol{w}}\right\|_2 \leq E \cdot T^{\frac{1}{4}}$ with large probability, therefore $\mathcal{L}$ is highly likely to be Lipschitz continuous with parameter $ET^{\frac{1}{4}}$. By setting a constant learning rate $\eta = \frac{DN}{E\sqrt{T}}$, and critical margin $\kappa_t = (1+\varepsilon)\log(\zeta T - t), \zeta > 1$, we have with probability at least $\left(1 - \frac{\alpha}{\mu T}\right)^T \geq 1 - \frac{\alpha}{\mu}$

$$\min_t \mathcal{L}(\boldsymbol{w}_t) - \mathcal{L}(\boldsymbol{w}^*) \leq \frac{DE}{NT^{\frac{1}{4}}} + \frac{D}{T}\sum_{t=1}^{T-1}\operatorname{Err}_t$$

$$\leq \frac{DE}{NT^{\frac{1}{4}}} + \frac{DE}{T^{\frac{3}{4}}}\sum_{t=1}^{T-1}\frac{1}{(\zeta T - t)^{1+\varepsilon}} + \frac{DE}{T^{\frac{3}{4}}}\sum_{t=1}^{T-1} r_t \tag{10}$$

$$\leq \frac{DE}{T^{\frac{1}{4}}}\left(\frac{1}{N} + \frac{c_{\varepsilon,\zeta}}{T^\varepsilon \sqrt{T}}\right) + \frac{DE}{T^{\frac{3}{4}}}\sum_{t=1}^{T-1} r_t$$

The first inequality follows the Theorem 1 in (Killamsetty et al., 2021). The last inequality holds because $\sum_{t=1}^{T-1}\frac{1}{(\zeta T - t)^{1+\varepsilon}} \leq \int_{(\zeta-1)T}^{\zeta T}\frac{1}{s^{1+\varepsilon}}ds \leq \frac{c_{\varepsilon,\zeta}}{T^\varepsilon}$ with $c_{\varepsilon,\zeta} = \frac{1}{\varepsilon(\zeta-1)^\varepsilon}, \forall \varepsilon > 0$ and $\zeta > 1$.

To bound the sum of classification error (the last term of Equation 10), again we utilize the data distribution prior. Note that the data points contribute to $r$ are quantified by the following set:

$$E = \{\boldsymbol{w}_o^\top \boldsymbol{x} > 0 \wedge \boldsymbol{w}^\top \boldsymbol{x} < -\kappa\} \cup \{\boldsymbol{w}_o^\top \boldsymbol{x} < 0 \wedge \boldsymbol{w}^\top \boldsymbol{x} > \kappa\} := E_1 \cup E_2$$

where $\boldsymbol{w}_o$ is the oracle classifier such that the true label is generated according to $y = \operatorname{sgn}(\boldsymbol{w}_o^\top \boldsymbol{x})$. Let $\phi$ represent the probability density function of standard Gaussian, we see that

$$
\begin{aligned}
r = \int_E \phi(x|\Sigma)dx &= 2\int_{E_1} \phi(x|\Sigma)dx \\
&\leq 2\int_{\{\boldsymbol{w}^\top \cdot x < -\kappa\}} \phi(x|\Sigma)dx = 2\Phi\left(-\frac{\kappa}{\sqrt{\boldsymbol{w}^\top \Sigma \boldsymbol{w}}}\right) \\
&\leq 2\Phi\left(-\frac{\kappa}{D\sqrt{\lambda}}\right)
\end{aligned}
$$

where $\lambda$ is the largest eigenvalue of $\Sigma$. Therefore, we have the following estimation:

$$
\begin{aligned}
\frac{1}{T^{\frac{3}{4}}} \sum_{t=1}^{T-1} r_t &\leq \frac{1}{T^{\frac{3}{4}}} \sum_{t=1}^{T-1} 2\Phi\left(-\frac{\kappa_t}{D\sqrt{\lambda}}\right) \\
&\leq \frac{2}{T^{\frac{3}{4}}} \sum_{t=1}^{T-1} \frac{\phi(\kappa_t/(D\sqrt{\lambda}))}{\kappa_t/(D\sqrt{\lambda})} \qquad \text{(Gaussian upper tail bound)} \\
&= \frac{2D\sqrt{\lambda}}{\sqrt{2\pi}(1+\varepsilon)} \frac{1}{T^{\frac{3}{4}}} \sum_{t=1}^{T-1} \frac{1}{\log(\zeta T - t)} e^{-\frac{(1+\varepsilon)^2}{2D^2\lambda} \log^2(\zeta T - t)} \\
&\leq \frac{2D\sqrt{\lambda}T^{\frac{1}{4}}}{\sqrt{2\pi}(1+\varepsilon)} \frac{1}{\log((\zeta-1)T+1)} \frac{1}{((\zeta-1)T+1)^{\frac{(1+\varepsilon)^2}{2D^2\lambda} \log((\zeta-1)T+1)}} \\
&\leq c_{\varepsilon,\zeta,\lambda} T^{-\beta}
\end{aligned}
\tag{11}
$$

where $\beta = \frac{(1+\varepsilon)^2}{2D^2\lambda} - \frac{1}{4}$ and we assume $\log((\zeta-1)T+1) = \Omega(1)$ with respect to $T$. Together we prove the theorem 2.2. $\qquad\square$

## C GENERALIZATION

Sorscher et al. (2022) analysed the generalization of static training scheme in the teacher-student perceptron setting, where the teacher is an "oracle" generating labels. For the training set $\mathcal{T} = \{\boldsymbol{x}_i, y_i\}_{i=1}^{|\mathcal{T}|}$, assume $\boldsymbol{x}_i \sim \mathcal{N}(0, \mathbf{I})$ and there exists an oracle model $\boldsymbol{w}_o \in \mathbb{R}^d$ which generates the labels such that $y_i = \operatorname{sign}(\boldsymbol{w}_o^\top \boldsymbol{x}_i)$ for all $i$. Without loss of generality, the oracle is assumed to be drawn form a sphere. Sorscher et al. (2022) works in a high dimensional statistics where $|\mathcal{T}|, d \to \infty$ but the ratio $\alpha = |\mathcal{T}|/d$ remains $\mathcal{O}(1)$.

Following the static training scheme, a lower fidelity estimator $\boldsymbol{w}_{\text{estimate}}$ which has angle $\theta$ relative to the oracle $\boldsymbol{w}_o$ is used to evaluate the candidate instances, and those with smaller classification margin $|\boldsymbol{w}_{\text{estimate}}^\top \boldsymbol{x}_i|$ along the estimator $\boldsymbol{w}_{\text{estimate}}$ are picked. The selection results in a subset $\mathcal{S}$. $\mathcal{S}$ follows $p(z)$, a truncated Gaussian distribution along $\boldsymbol{w}_{\text{estimate}}$, while the other directions are still kept isotropic. More specifically, given a keeping ratio $\gamma$, the corresponding selection margin is $\kappa = H^{-1}\left(\frac{1-\gamma}{2}\right)$ and thus the subset distribution along $\boldsymbol{w}_{estimate}$ is $p(z) = \frac{e^{-z^2/2}}{\sqrt{2\pi}\gamma}\Theta(\kappa - |z|)$, where $\Theta(x)$ is the Heaviside function and $H(x) = 1 - \Phi(x)$ where $\Phi(x)$ is the cumulative distribution function (CDF) of standard Gaussian.

The generalization error of the model trained on the subset $\mathcal{S}$ takes the form $\mathcal{E}(\alpha, \gamma, \theta)$. That is, the error is determined by $\gamma$ the keeping ratio, $\alpha$ which indicates the abundance of training samples before selection, and $\theta$ which shows the closeness of the estimator to the oracle model. The full set of self-consistent equations characterizing $\mathcal{E}(\alpha, \gamma, \theta)$ is given as

$$\frac{R - \rho\cos\theta}{\sin^2\theta} = \frac{\alpha}{\pi\Lambda}\left\langle \int_{-\infty}^{\nu} d\tau \exp\left(-\frac{\Delta(\tau,z)}{2\Lambda^2}\right)(\nu-\tau)\right\rangle_z$$

$$1 - \frac{\rho^2 + R^2 - 2\rho R\cos\theta}{\sin^2\theta} = 2\alpha\left\langle \int_{-\infty}^{\nu} d\tau \frac{e^{-\frac{(\tau-\rho z)^2}{2(1-\rho^2)}}}{\sqrt{2\pi}\sqrt{1-\rho^2}} H\left(\frac{\Gamma(\tau,z)}{\sqrt{1-\rho^2}\Lambda}\right)(\nu-t)^2\right\rangle_z$$

$$\frac{\rho - R\cos\theta}{\sin^2\theta} = 2\alpha\left\langle \int_{-\infty}^{\nu} d\tau \frac{e^{-\frac{(\tau-\rho z)^2}{2(1-\rho^2)}}}{\sqrt{2\pi}\sqrt{1-\rho^2}} H\left(\frac{\Gamma(\tau,z)}{\sqrt{1-\rho^2}\Lambda}\right)\left(\frac{z-\rho\tau}{1-\rho^2}\right)(\nu-\tau)\right.$$
$$\left. + \frac{1}{2\pi\Lambda}\exp\left(-\frac{\Delta(\tau,z)}{2\Lambda^2}\right)\left(\frac{\rho R - \cos\theta}{1-\rho^2}\right)(\nu-\tau)\right\rangle_z$$

$$(12)$$

Where,

$$\Lambda = \sqrt{\sin^2\theta - R^2 - \rho^2 + 2\rho R\cos\theta}$$
$$\Gamma(t,z) = z(\rho R - \cos\theta) - \tau(R - \rho\cos\theta)$$
$$\Delta(t,z) = z^2\left(\rho^2 + \cos^2\theta - 2\rho R\cos\theta\right) + 2\tau z(R\cos\theta - \rho) + \tau^2\sin^2\theta$$

$\tau$ is an auxiliary field introduced by Hubbard-Stratonovich transformation. $\langle\cdot\rangle_z$ denotes expectation on $p(z)$. By solving these equations the generalization error can be easily read off as $\mathcal{E} = \cos^{-1}(R)/\pi$, where $R = \frac{\boldsymbol{w}^\top\boldsymbol{w}_o}{\|\boldsymbol{w}\|_2 \cdot \|\boldsymbol{w}_o\|}$.

## D    MORE RESULTS ON GENERALIZATION

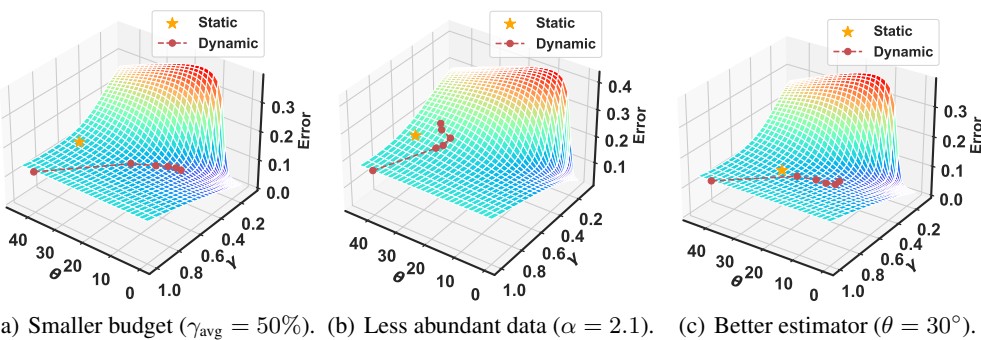

(a) Smaller budget ($\gamma_{\mathrm{avg}} = 50\%$). (b) Less abundant data ($\alpha = 2.1$). (c) Better estimator ($\theta = 30°$).

Figure 6: Effects of select ratio $\gamma_{\mathrm{avg}}$, initial data abundance $\alpha$ and the closeness of the estimator to the oracle model $\theta$ on the generalization.

To better understand the generalization under classification margin selection $\mathcal{E}(\alpha,\gamma,\theta)$, we provide more results to individually inspect the effect of (on average) select ratio $\gamma_{\mathrm{avg}}$, initial data abundance $\alpha$ and the closeness of the estimator to the oracle mode $\theta$. As shown in Figure 6(a), we changed $\gamma_{\mathrm{avg}}$ from $60\%$ to $50\%$, thus constructing a smaller selection budget case. In Figure 6(b), we use $\alpha = 2.1$ instead of $\alpha = 3.2$ to construct a less abundant data case, where the data before selection is insufficient. In Figure 6(c), we start selecting samples using a better estimator $\theta = 30°$ instead of $\theta = 40°$. All the other hyper-parameters aside from the inspected one are kept consistent to those used Figure 2(b), that is, $\gamma_{\mathrm{avg}} = 0.6$, $\alpha = 3.2$ and $\theta = 40°$. We see that with various $\gamma_{\mathrm{avg}}$ and $\theta$, DynaMS outperforms its static counterpart. The abundance of initial data, however, significantly

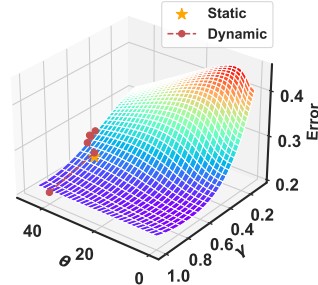

Figure 7: Generalization in a data scarce regime ($\alpha = 1.7$).

affects. When data is insufficient, data selection, both static as well as dynamic cause obvious performance degradation. Figure 7 shows a even more serious $\alpha = 1.7$, the generalization landscape is significantly changed and data selection is not recommended in this case.

## E  COMPARISON WITH STANDARD DEVIATION

We test each method in Table 4 and Table 5 5 times. The averaged accuracy and standard deviation are reported below in Table 6 and Table 7.

Table 6: Comparison for ResNet-18 on Cifar-10 with standard deviation.

| Methods | Types | Budget | Schedule | Accuracy. |
|---|---|---|---|---|
| Original | - | 100% | - | $95.52_{\pm 0.09}$ |
| Random | Stat. | 60% | - | $94.09_{\pm 0.23}$ |
| EL2N$_1$ | Stat. | 60% | - | $94.55_{\pm 0.15}$ |
| EL2N$_{10}$ | Stat. | 60% | - | $95.34_{\pm 0.13}$ |
| GraNd$_{10}$ | Stat. | 60% | - | $95.21_{\pm 0.15}$ |
| Forget$_{10}$ | Stat. | 60% | - | $95.29_{\pm 0.12}$ |
| OnlineMS | OLBS. | 60% | Const. | $95.21_{\pm 0.18}$ |
| Auto-assist | OLBS. | 60% | Const. | $92.37_{\pm 0.24}$ |
| DynaRandom | Dyna. | 60% | Linear | $94.45_{\pm 0.17}$ |
| DynaCE | Dyna. | 60% | Linear | $94.96_{\pm 0.21}$ |
| Craig | Dyna. | 60% | Const. | $94.36_{\pm 0.19}$ |
| GradMatch | Dyna. | 60% | Const. | $94.84_{\pm 0.17}$ |
| DynaMS | Dyna. | 60% | Linear | $95.28_{\pm 0.15}$ |

Table 7: Comparison for ResNet-18 and ResNet-50 on ImageNet with standard deviation

| | Types | Budget | Schedule | ResNet 18 | | ResNet 50 | |
|---|---|---|---|---|---|---|---|
| | | | | Top1 Acc. | Top5 Acc. | Top1 Acc. | Top5 Acc. |
| Original | - | 100% | - | $70.56_{\pm 0.04}$ | $89.95_{\pm 0.02}$ | $75.96_{\pm 0.04}$ | $92.75_{\pm 0.02}$ |
| Random | Stat. | 60% | - | $67.16_{\pm 0.10}$ | $87.50_{\pm 0.07}$ | $72.46_{\pm 0.10}$ | $90.85_{\pm 0.04}$ |
| EL2N$_1$ | Stat. | 60% | - | $66.38_{\pm 0.09}$ | $88.56_{\pm 0.05}$ | $72.03_{\pm 0.12}$ | $91.78_{\pm 0.04}$ |
| EL2N$_{10}$ | Stat. | 60% | - | $66.46_{\pm 0.10}$ | $88.73_{\pm 0.04}$ | $72.18_{\pm 0.10}$ | $92.02_{\pm 0.06}$ |
| GraNd$_{10}$ | Stat. | 60% | - | $66.50_{\pm 0.06}$ | $88.76_{\pm 0.03}$ | $72.14_{\pm 0.06}$ | $92.16_{\pm 0.03}$ |
| Forget$_{10}$ | Stat. | 60% | - | $67.84_{\pm 0.08}$ | $87.50_{\pm 0.03}$ | $73.50_{\pm 0.06}$ | $91.41_{\pm 0.04}$ |
| SVP+Forget | Stat. | 60% | - | - | - | $72.90_{\pm 0.10}$ | $91.37_{\pm 0.04}$ |
| SVP+Entropy | Stat. | 60% | - | - | - | $73.00_{\pm 0.01}$ | $91.52_{\pm 0.01}$ |
| DynaRandom | Dyna. | 60% | Power | $67.59_{\pm 0.05}$ | $87.62_{\pm 0.03}$ | $72.63_{\pm 0.12}$ | $90.91_{\pm 0.08}$ |
| DynaCE | Dyna. | 60% | Power | $67.58_{\pm 0.10}$ | $88.10_{\pm 0.04}$ | $72.80_{\pm 0.08}$ | $91.31_{\pm 0.03}$ |
| Craig | Dyna. | 60% | Const. | $65.32_{\pm 0.08}$ | $86.92_{\pm 0.04}$ | $70.69_{\pm 0.07}$ | $90.72_{\pm 0.02}$ |
| GradMatch | Dyna. | 60% | Const. | $66.48_{\pm 0.11}$ | $88.61_{\pm 0.04}$ | $71.79_{\pm 0.07}$ | $91.67_{\pm 0.03}$ |
| DynaMS | Dyna. | 60% | Linear | $68.12_{\pm 0.13}$ | $88.93_{\pm 0.06}$ | $74.10_{\pm 0.09}$ | $92.25_{\pm 0.03}$ |
| DynaMS | Dyna. | 60% | Power | $\mathbf{68.65_{\pm 0.11}}$ | $\mathbf{89.21_{\pm 0.04}}$ | $\mathbf{74.56_{\pm 0.09}}$ | $\mathbf{92.33_{\pm 0.03}}$ |
| DynaMS+PSP | Dyna. | 60% | Linear | - | - | $73.59_{\pm 0.09}$ | $91.79_{\pm 0.07}$ |
| DynaMS+PSP | Dyna. | 60% | Power | - | - | $73.40_{\pm 0.08}$ | $91.80_{\pm 0.01}$ |

## F  IMPLEMENTATION DETAILS AND HYPER-PARAMETERS

**Subset size schedule**  Dynamic Selection admits more freedom in subset size schedule. In the experiments we consider the *linear schedule* and the *power schedule*. For linear schedule, the keeping ratio is determined by $\gamma_k = 1 - k \cdot a$ for $k = 1, 2, \ldots, K$, where $a$ determines the sample reduction

ratio. $\gamma$ is supposed to satisfy $\gamma_{\mathrm{avg}} = \frac{1}{K} \sum_{k=1}^{K} \gamma_k = \gamma_s$ where $\gamma_s$ is the selection ratio when a static training scheme is applied. Thus $\frac{1}{K} \sum_{k=1}^{K} |\mathcal{T}_k| = |\mathcal{S}|$, meaning the averaged number of data used in the dynamic scheme is kept equal to that of static training.

Aside from the linear scheduler, we also explore a power schedule where $\gamma_k = m \cdot k^{-r} + b$ for $k = 1, 2, \ldots, K$. Power schedule reserves more samples in late training, preventing performance degradation caused by over data pruning. Determining these hyper-parameters $m, r, b$ is a bit tricky, we just require $\gamma_1 = 1.0$ to warm start and $\gamma_{\mathrm{avg}} = \frac{1}{K} \sum_{k=1}^{K} \gamma_k = \gamma_s$ for fair comparison. $\gamma_K$ should not be overly small, we empirically find $\gamma_K \approx \gamma - 0.1$ yield good results. For different budget $\gamma_s = \{0.6, 0.7, 0.8, 0.9\}$ the hyper-parameters are given in Appendix F, Table 8. Post process is carried out to make sure the resulting subset size sequence satisfy the above requirements.

(Killamsetty et al., 2021) utilize a *constant schedule*, where in each selection the subset size is kept constant as $\gamma_s \cdot |\mathcal{T}|$. This schedule however, do not admit selection without replacement. Linear and power schedule are all monotonically decreasing, thus are natural choices considering this. Figure 8 plots the three schedules on $\gamma_s = 0.6$ budget. In this paper we just provide a primary exploration on the subset size schedule, in depth study on the relationship between the subset size and the model performance as well as an automatic way determining the optimal subset size schedule is left for future work.

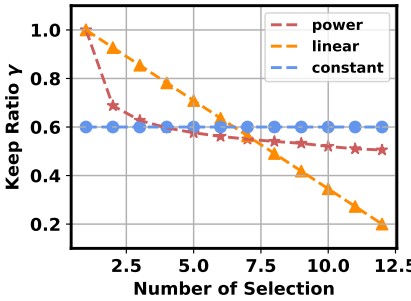

Figure 8: Different schedules for $\gamma_s = 0.6$ budget.

**Hyper-parameters**  Finally, the detailed hyper-parameters for DynaMS on both CIFAR-10 and ImageNet datasets are shown in Table 8. Note that for DynaMS+PSP, the Max Epochs is set to be 90 on ImageNet.

Table 8: Hyper-parameters of DynaMS for different models on CIFAR-10 and ImageNet.

| Hyper-parameters | CIFAR-10 | ImageNet | |
|---|---|---|---|
| | ResNet-18 | ResNet-18 | ResNet-50 |
| Batch Size | 128 | 512 | 512 |
| Init. Learning Rate of $\mathcal{W}$ | 0.1 | 0.1 | 0.1 |
| Learning Rate Decay | Stepwise 0.2 | Stepwise 0.1 | Stepwise 0.1 |
| Lr Decay milestones | $\{60,120,160\}$ | $\{40,80\}$ | $\{40,80\}$ |
| Optimizer | SGD | SGD | SGD |
| Momentum | 0.9 | 0.9 | 0.9 |
| Nestrov | True | True | True |
| Weight Decay | 5e-4 | 1e-4 | 1e-4 |
| Max Epochs | 200 | 120 | 120 |
| Selection interval | 10 | 10 | 10 |
| Power Scheduler | - | 60%: $m = 0.3984$, $r = 0.2371$, $b = 0.2895$
70%: $m = 0.3476$, $r = 0.2300$, $b = 0.4275$
80%: $m = 0.3532$, $r = 0.1349$, $b = 0.4978$
90%: $m = 0.2176$, $r = 0.1035$, $b = 0.7078$ | |
| Linear Scheduler | $a = 0.041$
-
-
- | 60%: $a = 0.073$
70%: $a = 0.055$
80%: $a = 0.036$
90%: $a = 0.018$ | |

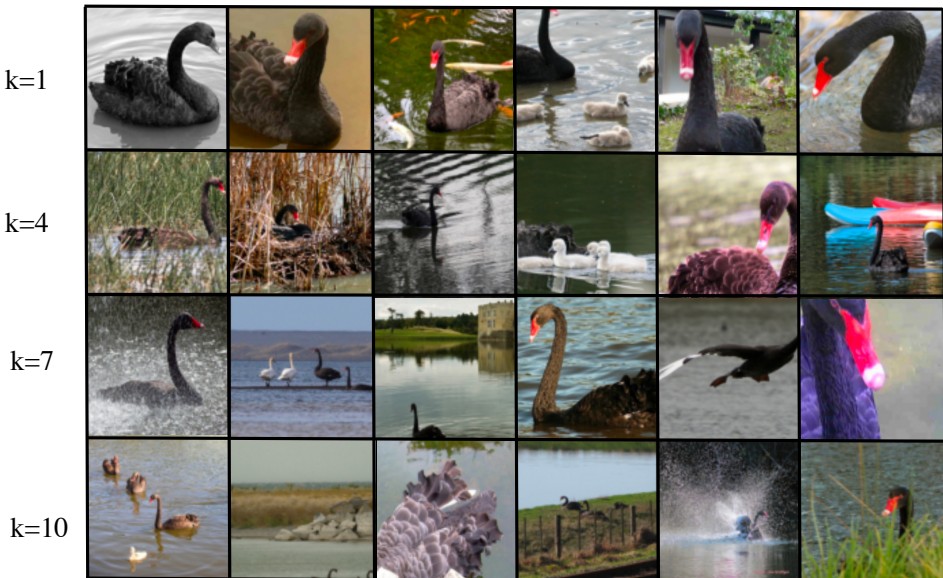

Figure 9: Images selected at different training stages of the model. As in (Sorscher et al., 2022), we show results on ImageNet class 100 (black swan).

## G    VISUALIZATION OF DYNAMICALLY SELECTED IMAGES

To get a better understanding of how the selected samples look like and how they change over time, we visualize samples picked in different selection steps along the training. For $k = 1, k = 4, k = 7$ and $k = 10$, which corresponds to the 1,4,7 and 10th selection, we randomly visualize selected samples that are absent in the latter selection. E.g. the $k = 4$ row shows images picked in the 4th selection but not in the 7th selection. From Figure 9, we see that in the early selections, amounts of easy-to-recognize samples are kept. As the training proceeds, these simple images are screened out and the model focuses more on harder samples that are atypical, blurred, or with interfering objects, validating our hypothesis that samples most informative change as the model evolves. Dynamic selection is thus indispensable.

# H    SUMMARY OF NOTATIONS

Table 9: Summary of the notations used throughout this paper. Variables only used in theoretical analysis including the convergence analysis and the generalization analysis are grayed for better readability.

| Topic | Notation | Explanation |
|---|---|---|
| Data (sub) Sets | $\mathcal{T}$ | The full training set |
| | $\lvert \cdot \rvert$ | Cardinality of a set |
| | $\boldsymbol{x}$ | data sample |
| | $y$ | data label |
| | $\mathcal{S}$ | The extracted subset |
| | $C$ | The number of classes |
| | $c$ | The $c$th class |
| Models and Parameters | $f(\cdot)$ | The model used for classification |
| | $\boldsymbol{w}$ | Parameters of the model |
| | $\boldsymbol{w}^*$ | Optimal model parameter |
| | $\boldsymbol{w}_o$ | Oracle model parameter |
| | $\boldsymbol{W}$ | Parameter of the linear classifier |
| | $\mathcal{W}$ | Kernel of a convolutional layers |
| | $\boldsymbol{g}$ | gradient incurred by the model |
| | $\boldsymbol{g}_{\text{proxy}}$ | gradient incurred by the proxy |
| | $d$ | The dimension of data feature |
| | $h(\cdot)$ | Feature extractor part of the model $f(\cdot)$ |
| | $p$ | Slimming factor, deciding the width of the proxy model |
| Selection schedule | $a$ | Sample reduction ratio in the linear schedule |
| | $m, r, b$ | Hyper-parameters controlling the power schedule |
| Loss Functions | $\mathcal{L}$ | Generic reference to the loss function |
| Data Selection | $\mathcal{B}$ | Decision boundary of linear classifiers |
| | $Q$ | Selection interval |
| | $M$ | The classification margin aka. distance of a sample to decision boundary |
| | $\gamma_k$ | Selection budget, keep ratio of samples for the $k$th selection |
| | $\gamma_{\text{avg}}$ | The averaged keep ratio of dynamic selection |
| | $\gamma_s$ | Selection budget in static selection. |
| | $k$ | Selection step |
| | $K$ | The total number of selections along training |
| | $\mathcal{E}$ | The generalization error of model trained on selected subset |
| | $\theta$ | Relative angle of a model to the oracle model. |
| | $\alpha$ | Aboundance of data before selection |
| | $\kappa$ | Selection margin. |
| Train | $t$ | Training epoch |
| | $T$ | The total number of training epochs, $T = Q \cdot (K+1)$ |
| Data Distribution | $\Sigma$ | Covariance of a Gaussian distribution |
| | $\lambda$ | The largest eigenvalue of the covariance matrix |
| Hyper-parameters | $D$ | Upper bound of model parameter norm |
| | $\varepsilon, \zeta, \mu$ | Constants appear in the convergence bound. |

