# OpenReview forum: "DynaMS: Dyanmic Margin Selection for Efficient Deep Learning"
_ICLR.cc/2023/Conference — ICLR 2023 poster_

### Official Review · Reviewer_UJJg · 2022-10-19

**Confidence:** 3
**Correctness:** 2
**Technical Novelty And Significance:** 2
**Empirical Novelty And Significance:** 2
**Recommendation:** 3

**Clarity, Quality, Novelty And Reproducibility:**

The paper is written in a good format and easy to follow.

Quality and Novelty are Fair.

Since there are some important equations and processes missing, I think this paper is hard to be reproduced.


**Strength And Weaknesses:**

Strength:
1. The topic of this paper is relatively new and can be used in many applications.
2. The performance of the proposed paper is strong.

Weakness:
1. The critical parts of this paper, the “SELECT” function and Power schedule, are missing. There are numerous hyperparameters in the supplementary, but no equation and process in the paper and supplementary.
2. “The Gaussian input assumption … infinitely a wide neural network resembles Gaussian process”. I think this assumption is against the research problem that uses smaller datasets and models. With limited number of data, the input will not follow Gaussian distribution.
3. The analysis on computational complexity is good but not enough. The comparison of the total training time between the proposed method and the original method from the ResNet paper is necessary.

Small issues:
1. In introduction, “redundant samples can be left out without sacrificing performance”. From 4, there is a clear drop in terms of performance, and Top 1 Acc. decreases over 1%.
2. The Top 1 Acc. results of “Original” in Table 4 and “Full” in Figure 4 are different.


**Summary Of The Paper:**

In this paper, the authors propose a new selection method to make train deep networks efficiently, which dynamically selects a small training
 set from the original data set. The selecting criterion is based on the classification margin. The authors also propose a lighter proxy model to speed up the selection process. The authors conduct experiments on CIFAR-10 and ImageNet to verify the effectiveness, and the results are over many baselines.

**Summary Of The Review:**

I do not recommend this paper now. If the authors can add missing parts, I will re-evaluate this paper.

---

> ### Author Response · Authors · 2022-11-18
> **Response to Reviewer UJJg**
>
> Thank you for your constructive comments. We address your concerns below:
>
> > The “SELECT” function and Power schedule, are missing. Numerous hyperparameters in the supplementary are not well introduced.
>
> Thanks for the suggestion. We originally described the "SELECT" operation in Section 2.1. We find it hard to depict the "SELECT" process with a single function. So, we add Algorithm 1 in Appendix A of the revision to depict the selection process, we hope this detailed description can make the selection procedure easier to understand. To better distinguish from other data selection strategies, we change the operator name "SELECT" to "MS", indicating the classification margin selection, in the revision.
>
> The power schedule is originally outlined in Appendix E (Appendix F in the revision),  the first line of paragraph 2. It takes the form $\gamma_k = m \cdot k^{-r} +b$, where $m,r,b$ are hyper-parameters given in Table 8 in Appendix E (Appendix F in the revision). As the reviewer suggested, we've moved it to the experiment part for completeness.
>
> As for the hyper-parameters, many of them are auxiliary variables used only in theoretical analysis. We gray them in Table 9 for better readability. In the revision, we made sure all the notations are properly introduced.
>
> > With limited number of data, the Gaussian distribution assumption is invalid in the convergence analysis.
>
> There might be a misunderstanding. Small data will not break the limit, the predictions of a neural network throughout gradient descent training resembles a GP as long as the model width goes to infinity. The Gaussian nature of the output is a result of Gaussian initialization and infinite width. The data size does not change this property, but only affects the covariance matrix. Please refer to Theorem 2.2 of [1] for more information.
>
> > More detailed training time analysis.
>
> Thank you for this suggestion, which is critical. We've added time and space complexity analysis of different selection strategies in Table 2 as well as detailed running time comparisons in Table 4 and Table 5. Please kindly refer to the general response for additional discussions on this question.
>
> > The Top 1 Acc. results of “Original” in Table 4 and “Full” in Figure 4 are different.
>
> Thanks for the careful checking. There exists a difference because the best full model accuracy is reported in Figure 5 while the averaged full model accuracy is reported in Table 4. In the revision, we resort to the averaged full accuracy in Figure 5.
>
> > Since there are some important equations and processes missing, I think this paper is hard to be reproduced.
>
> We would like to do our best to clearly present our work and improve the paper. We have added Algorithm 1 to detail the selection procedure and give more description on the power schedule thanks to your suggestion. We hope these will make the approach more clear and easy to reproduce as reviewer hmw5 and 6uhW stated. The code will be released upon acceptance.
>
> We thank the reviewer again for the review. If you find our added material useful, we respectfully ask that you please consider increasing the score. If there are still other equations and processes missing, or any follow-up questions and concerns, please don't hesitate to let us know, we would be happy to clear them and keep this discussion going.
>
> ---------------------
>
> [1] J. Lee et.al., Wide neural networks of any depth evolve as linear models under gradient descent. NeurIPS'19.

---

> ### Author Response · Authors · 2022-11-30
> **Thank you again for your review.**
>
> Again, thank you for the valuable feedback. Please let us know if you have any other questions/suggestions.

---

### Official Review · Reviewer_6uhW · 2022-10-23

**Confidence:** 4
**Correctness:** 3
**Technical Novelty And Significance:** 3
**Empirical Novelty And Significance:** 3
**Recommendation:** 3

**Clarity, Quality, Novelty And Reproducibility:**

Over, the paper is clearly written. The quality is fair. The idea of dynamic selection is novel. Also, the paper seems easy to reproduce.


**Strength And Weaknesses:**

Strength:

1. Reducing sample complexity is an important topic in deep learning. The authors propose some interesting ideas to address this problem.
2. Overall, the paper is clearly written and the proposed idea is intuitive.
3. Extensive ablation studies are conducted to show the benefit of the proposed approach.

Weaknesses:
1. The use cases of the proposed approach is not clear. If all the training samples are available for dynamic selection, then why not just use all the samples? Also, the authors did not show results of using all the samples. If sample selection is for computational complexity, there is also no time comparison.

2. Compared with dataset distillation, does the proposed approach have benefits in terms of accuracy and complexity?

3. It is not clear which samples are selected and how the samples selected will change over time.

4. Several typos:

i. "Our proposed DynaMS form" -> "Our proposed DynaMS forms"
ii. "is keep fixed" -> "is kept fixed"
iii. "as the model evolve" -> "as the model evolves"

**Summary Of The Paper:**

In this paper, the authors propose a subset selection method for training DNNs. The idea is to select the samples that are close to the margin dynamically. To reduce computation, the authors also propose Parameter Sharing Policy for sample selection. The authors show that the proposed DynaMS can converge tot the optimal solution with large probability. Experiments are conducted on CIFAR10 and ImageNet to show the effectiveness of the proposed approach.

**Summary Of The Review:**

The authors propose a dynamic sample selection method for DNN training and conduct extensive experiments to show the benefits. However, there are several issues with the current draft as stated in the drawbacks.

---

> ### Author Response · Authors · 2022-11-18
> **Response to Reviewer 6uhW**
>
> Thank you for your constructive comments. We address your concerns below:
>
> > The use cases of the proposed approach is not clear.
>
> The main purpose of this paper is to accomplish efficient training with the most informative subset, that is, to reduce the training time computational complexity. The problem has already been explored in previous works like [1,2,3] and is well-defined. The results of using all the samples are given in the first line of Table 3 and Table 4. In the revision, we conducted extensive comparisons on the run-time of different methods. Please kindly refer to the general response for additional discussions on this question.
>
> > Compare with dataset distillation.
>
> Thanks for the reminder, we made up a missed lesson on the dataset distillation literature[4,5,6,7] and find that this technique is significantly different from ours.
>
> The goal is not the same. Dataset distillation aims to achieve very aggressive compression (storage) e.g. 10 or 50 samples per class while we aim at accelerating the training process (time). Typically, dataset distillation is time-consuming, [4] requires second-order derivatives to generate the synthesized data points. As a consequence of the extremely high compression rate, data distillation suffers from higher accuracy drop (over 15\% for CIFAR-10 in [7]) while we try to maximally retain the model utility (within 0.5\% for CIFAR-10 ).
>
> Besides, as far as we know, the dataset distillation technique is still limited to small datasets like MNIST, CIFAR-10, and CIFAR-100. It hasn't been successfully applied to large-scale datasets like ImageNet. Please refer to the "paper decision" thread of [7] https://openreview.net/forum?id=hXWPpJedrVP.
>
> > Not clear which samples are selected and how the samples selected will change over time.
>
> Thank you for this constructive comment. We've added an appendix visualizing samples picked in different selections in Appendix G, which we believe will give a better understanding on how will the most informative samples change along the training process. From Figure 9 in Appendix G, we see that in the early selections, amounts of easy-to-recognize samples are kept. As the training proceeds, these simple images are screened out and the model focus more on harder samples that are atypical, blurred, or with interfering objects, validating our hypothesis that samples most informative change as the model evolves. Dynamic selection is thus indispensable.
>
> >  Several typos.
>
> Thanks very much for your careful checking, we've fixed these typos and thoroughly polished the revision.
>
> If you found our clarification helpful, we respectfully ask that you consider increasing the score. Please let us know if you have any follow-up questions. We are happy to keep this discussion going.
>
> ----------------------------
>
> [1] J. Zhang et.al. AutoAssist: A Framework to Accelerate Training of Deep Neural Networks. NeurIPS'19.
>
> [2] B. Mirzasoleiman et.al. Coresets for Data-efficient Training of Machine Learning Models. ICML'20.
>
> [3] K. Killamsetty et.al. GRAD-MATCH: Gradient Matching based Data Subset Selection for Efficient Deep Model Training. ICML'21
>
> [4] T.Wang et.al. Dataset Distillation. arXiv preprint, arXiv:1811.10959.
>
> [5] B.Zhao et.al. Dataset Condensation with Gradient Matching. ICLR’21
>
> [6] B.Zhao et.al. Dataset Condensation with Differentiable Siamese Augmentation. ICML'22
>
> [7] T. Nguyen et.al. Dataset Distillation with Infinitely Wide Convolutional Networks. NeurIPS'21.

---

> ### Author Response · Authors · 2022-11-30
> **Thank you again for your review.**
>
> Again, thank you for the valuable feedback. Please let us know if you have any other questions/suggestions.

---

### Official Review · Reviewer_hmw5 · 2022-10-24

**Confidence:** 4
**Correctness:** 3
**Technical Novelty And Significance:** 2
**Empirical Novelty And Significance:** 2
**Recommendation:** 6

**Clarity, Quality, Novelty And Reproducibility:**

This paper is presented well and the idea of dynamic selection based on class margin is natural and reasonable. The whole pipeline is novel and easy to reproduce with the detailed implementation.

**Strength And Weaknesses:**

Strength:
1. Selecting informative subset by class margin is reasonable, and the generalization of DynaMS has been well supported both in practice and theory.
2. The performance is superior compared with previous SOTA methods.
3. This paper is well written and the implementation is described in detail which is easy to follow.

Weaknesses:
1. The need for PSP is not significant. As mentioned in this paper, the selection procedure is efficient and its overhead is negligible since it is conducted only 19 times during training (200 epochs in total).
2. Training slimming networks is not efficient because multiple sub-networks are trained separately. For example, compared to the original training time, with a 60% budget, the overall training cost of DynaMS+PSP is $0.6\times (1+\frac{9}{16}+\frac{1}{4}+\frac{1}{16})>1$, while other methods almost strictly cost 0.6 of the original one.
3. For the results shown in Figure 2(a), only 10 epoch training on one dataset is not convincing enough. Why not report overlap ratio of all evaluated epochs on both CIFAR-10 and ImageNet?

**Summary Of The Paper:**

This paper proposes to dynamically select partial training data for efficient learning. The main idea is updating informative subset according to their margin to class decision boundary. A parameter sharing proxy strategy is devised to further evaluate instance prior. As a result, the proposed method achieves superior performance compared with other SOTA data selection strategies on different budgets.

**Summary Of The Review:**

This paper present a good idea to select informative subset for efficient training. The proposed method achieves superior performance and generalization. However, the PSP module seems redundant for the whole pipeline and it weakens the contribution based on my current understanding. I am glad to change my score if my concerns are well addressed.

---

> ### Author Response · Authors · 2022-11-18
> **Response to Reviewer hmw5**
>
> We sincerely appreciate your recognition of our technical contributions. We address your concerns below:
>
> > The need for PSP is not significant.
>
> PSP is especially advantageous for hard, extremely large-scale problems, where selection is conducted frequently and scanning all the candidate samples with the underlying model is unaffordable. This is the case for many real-world applications like modern foundation model training. That is, the design of PSP is not confined to the current experiment setting. As the contribution of the PSP is a common concern of reviewers, please kindly refer to the general response for additional discussions on this question.
>
> > Training slimming networks is not efficient.
>
> We agree that training a slimmable network for one epoch takes more time than conventional training. But still, we can make DynaMS+PSP efficient. In this paper, we use only a $\times0.5$ proxy instead of training {$\times0.25, \times0.5, \times0.75$} sub-models simultaneously as in [1], so the overall one-epoch training cost of DynaMS+PSP is 0.6$\times(1 + \frac{1}{4})$. However, as we have shown in Figure 2(c), the gradient of the proxy and the original model is well aligned so the slimmable model requires fewer training epochs to converge. In our experiments, we train for 90 epochs instead of 120 as the original model, so the overall theoretical training time is $\frac{3}{4}\times0.6\times(1 + \frac{1}{4})=0.5625$, which is no more expensive than other methods. Note that in Section 4.2, paragraph "The effect of parameter sharing proxy", we just launch $\{\times0.25, \times0.5, \times0.75\}$ proxies to ablate study the effects of different proxy widths. in practice, we use only one proxy.
>
> > The results shown in Figure 2(a), only 10 epoch training on one dataset is not convincing enough.
>
> There might be a misunderstanding. The overlap ratio shown in Figure 2(a) is actually the overlap ratio all through the whole training process instead of only 10 epochs. For ImageNet experiment, we train the model for 120 epochs and conduct selection after 10 epochs' training, so there are overall 11 times selection and altogether 10 overlap ratio values. We've made this more clear in the revision and added the CIFAR-10 results.
> On CIFAR-10 we train for 200 epochs and also conduct selection every 10 epochs. We see that on both CIFAR-10 and ImageNet, the overlap ratio is lower than 1.0, validating our hypotheses that the most informative subset varies as the model evolves. In our experiment, CIFAR-10 got on average 0.83 overlap ratio, which is higher than the 0.73 for ImageNet. This partly explains why static methods can also achieve good results on simple datasets like CIFAR-10 but degenerate on ImageNet.
>
> We thank the reviewer again for the review. If you have any more questions about our replies, or any further questions or concerns, please let us know.
>
> -----------------------------
>
> [1] J. Yu et.al. Slimmable Neural Networks. ICLR'19.

---

> > ### Comment · Reviewer_hmw5 · 2022-12-01
> > **Response to the effect of PSP**
> >
> > Thanks for the responses. The efficiency issue of PSP has been addressed. However, this paper does not verify PSP for the desired experiment setting, i.e., hard and large-scale problems, and the current effect is not significant (12.8 hr vs 13.0 hr). Considering the main success of the selection strategy, I keep my score. Thanks.

---

> ### Author Response · Authors · 2022-11-30
> **Thank you again for your review.**
>
> Again, thank you for the valuable feedback. Please let us know if you have any other questions/suggestions.

---

### Official Review · Reviewer_8YeC · 2022-10-25

**Confidence:** 4
**Clarity, Quality, Novelty And Reproducibility:** The quality and clarity is good, and …
**Correctness:** 3
**Technical Novelty And Significance:** 2
**Empirical Novelty And Significance:** 2
**Recommendation:** 5

**Strength And Weaknesses:**

Strength：
+ A dynamic margin selection (DynaMS) method is proposed to dynamically construct the training subset by utilizing the distance from candidate samples to the classification boundary.
+ Extensive analysis and experiments are conducted to show the performance of the proposed method.

Weaknesses：
- The paper claims that the existing sample selection methods are expensive both in run-time and in memory efficiency, however, the paper lacks comparisons with different methods in run-time and memory.
- In Table 3, the proposed DynaMS does not perform better than Stat.-based methods, and performs slightly better than Dyna.-based method such as CRAIG and GradMatch.
- In Table 4, the results of DynaMS+PSP are not competitive, so the contribution of PSP is limited.

**Summary Of The Paper:**

This paper introduces a dynamic margin selection (DynaMS) method to dynamically construct the training subset by utilizing the distance from candidate samples to the classification boundary. In addition, a light parameter sharing proxy is designed to reduce the additional computation incurred by the selection. Extensive analysis and experiments demonstrate the superiority of the proposed approach in data selection.

**Summary Of The Review:**

The major concerns of this work are as
- The paper claims that the existing sample selection methods are expensive both in run-time and in memory efficiency, however, the paper lacks comparisons with different methods in run-time and memory.
- In Table 3, the proposed DynaMS does not perform better than Stat.-based methods, and performs slightly better than Dyna.-based method such as CRAIG and GradMatch.
- In Table 4, the results of DynaMS+PSP are not competitive, so the contribution of PSP is limited.

---

> ### Author Response · Authors · 2022-11-18
> **Response to Reviewer 8YeC**
>
> Thank you for your constructive comments. We address your concerns below:
>
> > Lacks comparisons with different methods in run-time and memory.
>
> We agree this is a very important point and we have added detailed comparisons along run-time and memory in the revision. As this is a common concern of many reviewers, please kindly refer to the general response for additional discussions on this question.
>
> > DynaMS does not perform better than static-based methods on small data CIFAR-10.
>
> In the experiments, we find that CIFAR-10 is relatively easy and most of the approaches reach comparable accuracy as training on the full dataset. So we focus on the more challenging ImageNet, where DynaMS outperform other baselines significantly. Besides, even on CIAFR-10, DynaMS outperforms the static methods in efficiency. Under 0.6$\times$ budget, DynaMS accomplishes training in 21.1 minutes while well-performing static methods take at least 46.9 minutes taking into account the pretraining and selection cost. Please refer to Table 4 for more information.
>
> >  DynaMS+PSP is not competitive, so the contribution of PSP is limited.
>
> Conceptually, PSP is the only proxy suitable for dynamic selection, which is essential for maximally maintaining the model utility. Practically, PSP is advantageous for extremely large-scale problems, where scanning all the candidate samples with a lighter proxy can
> considerably save time. This is true for modern foundation model training. As the contribution of the PSP is a common concern of reviewers, please kindly refer to the general response for additional discussions on this question.
>
> As for the performance of DynaMS+PSP, we find that utilizing proxies inevitably introduces bias, leading to slight performance degradation. Nevertheless, DynaMS+PSP still outperforms all the other baselines, especially SVP, which incorporates a statically pre-trained proxy model. The superiority of DynaMS+PSP over SVP shows the necessity of a dynamic proxy that agilely keeps up with the change of the underlying model. Note that currently the proxy is constructed roughly by uniformly scaling down the original model, which is sub-optimal. By cleverly manipulating the width of each layer[1,2], the performance of the DynaMS+PSP can be further improved.
>
> Since we have replied to all the questions, if you find our answers satisfactory, we respectfully ask that you please consider increasing the score. If you have any more questions or concerns, please let us know and we will be happy to answer.
>
> ------------------------------------
>
> [1] Y. He et.al. AMC: AutoML for Model Compression and Acceleration on Mobile Devices. ECCV'18
>
> [2] X. Dong and Y. Yang, Network Pruning via Transformable Architecture Search. NeurIPS'19

---

> ### Author Response · Authors · 2022-11-30
> **Thank you again for your review.**
>
> Again, thank you for the valuable feedback. Please let us know if you have any other questions/suggestions.

---

### Author Response · Authors · 2022-11-18
**Response to all reviewers (Part1)**

Dear Reviewers,

We sincerely thank all reviewers for the constructive comments. For the comments and concerns in reviews, we first write a general response to address the major ones. Then we separately respond to each individual question. We have also revised the paper to accommodate all the suggestions and improve the presentation as well.

The major concerns from reviewers are on:
1. The use cases and the value of the proposed approach (Reviewer 6uhW).
2. Comparisons with different methods in run-time and memory (Reviewer 8YeC, Reviewer 6uhW, and Reviewer UJJg).
3. The contribution of PSP (Reviewer 8YeC and Reviewer hmw5 ).

----------------------
We address each below.

### 1. The use case and the value of the approach (Reviewer 6uhW).
Our primary goal is to select informative subsets for efficient training, aka, accelerate training. A detailed comparison with different methods on time and space complexity as well as the actual running time is given in the general response to the next question.
Here we also highlight the value of this approach. **In practice,** all the reviewers reach the consensus that DynaMS achieves superior performance compared with previous methods, showing its practical value. **In theory,** we conduct convergence analysis as well as generalization analysis for the proposed DynaMS. To the best of our knowledge, this is ***the first time*** proved that with proper assumptions, select samples according to classification margin converges with high probability, and dynamically conducting selection can improve the generalization upon its conventional static counterparts.

### 2. Comparisons with Different methods in run-time and memory (Reviewer 8YeC, Reviewer 6uhW, and Reviewer UJJg).
Thanks for this suggestion, which is a very important point. We first conduct a computational and space complexity analysis contrasting different selection strategies. The complexity is as follows:

Strategy| Time Complexity |  Space Complexity
-|-|-
CE-loss | $\mathcal{O}((C + \log \|\mathcal{S}\|) \cdot \|\mathcal{T}\|)$ | $\mathcal{O}(\|\mathcal{T}\|)$
EL2N    | $\mathcal{O}((C + \log \|\mathcal{S}\|) \cdot \|\mathcal{T}\|)$  | $\mathcal{O}(\|\mathcal{T}\|)$
GraNd  | $\mathcal{O}((C \cdot d + \log \|\mathcal{S}\|) \cdot \|\mathcal{T}\|)$ | $\mathcal{O}(\|\mathcal{T}\|)$
Craig    | $\mathcal{O}(C \cdot d \cdot \|\mathcal{T}\| \cdot \|\mathcal{S}\|)$ | $\mathcal{O}(\|\mathcal{S}\| \cdot \|\mathcal{T}\|) $
GradMatch |  $\mathcal{O}(C \cdot d \cdot \|\mathcal{T}\| \cdot \|\mathcal{S}\|)$  |  $\mathcal{O}(C \cdot d \cdot \|\mathcal{T}\|)$
MS | $\mathcal{O}(C \cdot (d + \log \|\mathcal{S}\|) \cdot \|\mathcal{T}\|)$| $\mathcal{O}(C \cdot \|\mathcal{T}\|)$

Here $C$ indicates the number of classes, $|\mathcal{T}|$ is the original dataset size and $|\mathcal{S}|$ is the selected subset size. $d$ is the dimension of the data feature. We see that classification margin selection (MS) is slightly slower than selection via loss (CE-loss and EL2N), but much more efficient than Craig and GradMatch. Here we consider only the complexity of the selection strategy itself, time spent for feature extraction and model training is not included.

We also report the actual running time. On CIFAR-10, the run-time comparison is given in the following table:

Method        |   Type   | Budget    | Schedule | Acc$_{\pm \text{std}}$ |  Time(min)
-|-|-|-|-|-
Original            |  - | 100% |  - | 95.52$_{\pm \text{0.09}}$ | 28.6
Random          | Static  | 60%   |  - | 94.09$_{\pm \text{0.23}}$ | 17.4
EL2N$_1$        |  Static  | 60% |  - | 94.55$_{\pm \text{0.15}}$ | 17.4$_{+2.9}$
EL2N$_{10}$    |  Static | 60%  |  - | 95.34$_{\pm \text{0.13}}$ | 17.4$_{+29.5}$
GraNd$_{10}$  |  Static  |  60% | -| 95.21$_{\pm \text{0.15}}$  |  17.4$_{+29.5}$
Forget$_{10}$   |  Static  | 60% | - | 95.29$_{\pm \text{0.12}}$  |  17.4$_{+88.5}$
OnlineMS          | OLBS  |  60% | Const. | 95.21$_{\pm \text{0.18}}$ | 9076.3
Auto-assist       |  OLBS   | 60% | Const. |  92.37$_{\pm \text{0.24}}$| 24.6
DynaRandom   |  Dynamic  | 60% | Linear | 94.45$_{\pm \text{0.17}}$ | 20.9
DynaCE            |  Dynamic   | 60%  | Linear | 94.96$_{\pm \text{0.21}}$  | 21.0
Craig                |  Dynamic   |  60% | Const. | 94.36$_{\pm \text{0.19}}$  |  28.7
GradMatch      |  Dynamic    | 60%  | Const. |  94.84$_{\pm \text{0.17}}$  |  26.7
DynaMS           | Dynamic   |  60%  | Linear | 95.28$_{\pm \text{0.15}}$  | 21.3

We see that DynaMS achieves comparable performance against the strongest baselines (EL2N10, GraNd10, Forget10) while being more efficient. Note that all the static methods aside from Random require pretraining one or several models for 20 epochs before subset selection. Considering this cost(subscript of the reported running time), the overall acceleration of the static methods is less significant.

---

> ### Author Response · Authors · 2022-11-18
> **Response to all reviewers (Part2)**
>
> On ImageNet, the run-time comparison of ResNet-18 and ResNet-50 is given in the following tables:
>
> Method             |    Type   |     Budget   | Schedule |  Acc. Top 1 |  Acc. Top 5  | Time(hrs)
> -|-|-|-|-|-|-
> Original            |  -        | 100% | - |  70.56$_{\pm \text{0.04}}$ | 89.95$_{\pm \text{0.02}}$  |   12.8
> Random          | Static  | 60%   | - | 67.16$_{\pm \text{0.10}}$ | 87.50$_{\pm \text{0.07}}$ |  7.6
> EL2N$_1$        |  Static  | 60%   | - |  66.38$_{\pm \text{0.09}}$ |  88.56$_{\pm \text{0.05}}$ |   7.6$_{+2.1}$
> EL2N$_{10}$    |  Static   | 60%   | - | 66.46$_{\pm \text{0.10}}$ |  88.73$_{\pm \text{0.04}}$  |   7.6$_{+21.1}$
> GraNd$_{10}$  |  Static  |  60%  | - | 66.50$_{\pm \text{0.06}}$  |  88.76$_{\pm \text{0.03}}$ |    7.6$_{+21.1}$
> Forget$_{10}$   |  Static  |  60%  | - | 67.84$_{\pm \text{0.08}}$  |  87.50$_{\pm \text{0.03}}$ |   7.6$_{+63.1}$
> DynaRandom   |  Dynamic  | 60%  | Power|  67.59$_{\pm \text{0.05}}$ | 87.62$_{\pm \text{0.03}}$  |  8.9
> DynaCE            |  Dynamic   | 60%   | Power | 67.58$_{\pm \text{0.10}}$  | 88.10$_{\pm \text{0.04}}$  |   9.2
> Craig                |  Dynamic   |  60%  | Const. | 65.32$_{\pm \text{0.08}}$  |  86.92$_{\pm \text{0.04}}$  |  12.1
> GradMatch      |  Dynamic    | 60%   | Const. | 66.48$_{\pm \text{0.11}}$  | 88.61$_{\pm \text{0.04}}$ |  11.7
> DynaMS           | Dynamic   |  60% | Linear | 68.12$_{\pm \text{0.13}}$  | 88.93$_{\pm \text{0.06}}$  |  9.6
> DynaMS           | Dynamic   |  60% | Power |  **68.65$_{\pm \text{0.11}}$**  | **89.21$_{\pm \text{0.04}}$**  | 9.6
>
>
> Method             |    Type   |     Budget   | Schedule | Acc. Top 1 |  Acc. Top 5  | Time(hrs)
> -|-|-|-|-|-|-
> Original            |  -        | 100% | - |  75.96$_{\pm \text{0.04}}$ | 92.75$_{\pm \text{0.02}}$  |   18.2
> Random          | Static  | 60%   | - |  72.46$_{\pm \text{0.10}}$ | 90.85$_{\pm \text{0.04}}$ |   10.9
> EL2N$_1$        |  Static  | 60%   | - |  72.03$_{\pm \text{0.12}}$ | 91.78$_{\pm \text{0.04}}$ |  10.9$_{+3.0}$
> EL2N$_{10}$    |  Static   | 60%   | - |  72.18$_{\pm \text{0.10}}$ | 92.02$_{\pm \text{0.06}}$  |  10.9$_{+29.7}$
> GraNd$_{10}$  |  Static  |  60%  | - |  72.14$_{\pm \text{0.06}}$  |  92.16$_{\pm \text{0.03}}$ |  10.9$_{+29.7}$
> Forget$_{10}$   |  Static  |  60%  | - |  73.50$_{\pm \text{0.06}}$  |  91.41$_{\pm \text{0.04}}$  |10.9$_{+89.1}$
> SVP+Forget      |  Static  |  60%  | - |  72.90$_{\pm \text{0.10}}$  |  91.37$_{\pm \text{0.04}}$ | 10.9$_{+12.8}$
> SVP+Entropy    |  Static  |  60%  | - |  73.00$_{\pm \text{0.01}}$  |  91.52$_{\pm \text{0.01}}$ | 10.9$_{+12.8}$
> DynaRandom   |  Dynamic  | 60%  | Power | 72.63$_{\pm \text{0.12}}$ | 90.91$_{\pm \text{0.08}}$   |    12.1
> DynaCE            |  Dynamic   | 60%   | Power | 72.80$_{\pm \text{0.08}}$  | 91.31$_{\pm \text{0.03}}$  |   12.5
> Craig                |  Dynamic   |  60%  | Const. | 70.69$_{\pm \text{0.07}}$  |  90.72$_{\pm \text{0.02}}$  |   16.0
> GradMatch      |  Dynamic    | 60%   | Const. | 71.79$_{\pm \text{0.07}}$  |  91.67$_{\pm \text{0.03}}$ |     15.2
> DynaMS           | Dynamic   |  60% |  Linear | 74.10$_{\pm \text{0.09}}$  | 92.25$_{\pm \text{0.03}}$    |    12.9
> DynaMS           | Dynamic   |  60% |  Power | **74.56$_{\pm \text{0.09}}$**  | **92.33$_{\pm \text{0.03}}$**   |   13.0
> DynaMS+PSP  | Dynamic   |  60% |  Linear | 73.59$_{\pm \text{0.09}}$   | 91.79$_{\pm \text{0.07}}$    |  12.8
> DynaMS+PSP  | Dynamic   |  60% |  Power | 73.40$_{\pm \text{0.08}}$  | 91.80$_{\pm \text{0.01}}$ |   12.8
>
> We see that DynaMS outperforms all the baselines.

---

> > ### Author Response · Authors · 2022-11-18
> > **Response to all reviewers (Part3)**
> >
> > ### 3. The contribution of PSP (Reviewer 8YeC and Reviewer hmw5).
> > We outline the contribution of PSP with the following two views.
> >
> > **Conceptually:**  Utilizing a small proxy to efficiently evaluate instances has been a common practice[1,2,3]. These proxies, however, are either static[1], insufficient[2] (Using a single perceptron layer as a proxy is insufficient for instance evaluation), or inefficient[3]. PSP is the first proxy that fits the requirement of dynamic selection, which we have shown to be critical to maximally retrain performance in data selection. It can agilely keep up with the model training dynamic, is expressive enough to evaluate the candidate instances, and is efficient due to the well-aligned gradient of the proxy and the underlying model.
> >
> > **Practically:** PSP can be significant in real-world applications, especially for large-scale, hard problems. When the data is extremely large that only a small fraction of samples is sufficient, training PSP on the subset can be cheaper than evaluating the whole large training set with the original model. When the task is hard and the model changes rapidly during training, PSP timely updates the informative subset, maximally retaining the model utility.  These advantages are important as data selection is most demanding in extremely large-scale problems like foundation model training.
> >
> > We thank all the reviewers for their constructive feedback. Hope the concerns are well addressed. We welcome additional comments.
> >
> > ---------------------------
> >
> > [1] C. Coleman et.al. Selection via Proxy: Efficient Data Selection for Deep Learning. ICLR'20.
> >
> > [2] J. Zhang ei.al. AutoAssist: A Framework to Accelerate Training of Deep Neural Networks. NeurIPS'19.
> >
> > [3] J. Yoon et.al. Data Valuation using Reinforcement Learning. ICML'20.

---

### Author Response · Authors · 2022-11-22
**Corrention of a few  typos and vague descriptions.**

Dear reviewers,

We find a few typos and vague descriptions in Algorithms 2 and 3 in the Appendix. We correct them here and hope this correction can help reduce confusion.

1. In Algorithm 2, in the third line of  **Input**, it should be $\gamma_k$ instead of $\gamma_t$.

2. In Algorithm 3, in the third line of **Input**, it should be $\gamma_k$ instead of $f_k$.

3. In line 7 of  Algorithms 2 and 3, the statement "Keep subset $\mathcal{S}_k=$

    $ \mathcal{S}_{k-1}$" is not precise, we change it to "Keep subset $\mathcal{S}_k$".

Please let us know if there are any more vague descriptions or follow-up questions.

---

### Comment · Area_Chair_ddi4 · 2022-12-07
**Response to Author Feedback**

Dear Reviewers, thank you so much again for your time on this paper. Thank you also Reviewer hmw5 for responding to the author's feedback. Other reviewers: the discussion phase is still ongoing, how does the author response and other reviews change your view of the paper?

---

### Decision · Program_Chairs · 2023-01-20

**Decision:**

Accept: poster

**Justification For Why Not Higher Score:**

The contribution of the work, while useful, is not ground-breaking.

**Justification For Why Not Lower Score:**

The work has all the hallmarks of a useful contribution to the community: it is simple, and works in a wide-variety of settings. Accepting it would improve ICLR.

**Metareview: Summary, Strengths And Weaknesses:**

The reviewers were split about this paper and did not come to a consensus: on one hand they appreciated the experimental analysis, on the other they had doubts about the use cases of the method and the clarity of the writing. After going through the paper and the discussion I have decided to vote to accept for the following reason: the main arguments against accepting are 1) Critical descriptions of the paper are missing; 2) The assumption of Gaussian inputs is unrealistic; 3) More detailed computation analysis is missing; 4) The usecase is unclear. However, the authors respond to each one of these points convincingly. For 1), the descriptions in question actually do appear in the paper, one in Section 2.1 another in the Appendix. For 2), the main complaint was that the assumption would not hold for small models or small data. However, the first point, while true, has no bearing on this work as it is aimed at reducing training time for large models. For models with large widths (i.e, > 50) G de G. Matthews et al., 2018 show that certain network outputs do converge (exponentially) to a Gaussian process. The second point actually does not affect the theoretical result (and is again not the setting considered in the paper). For 3) the authors have added additional time and space complexity analyses and running time comparisons. For 4) a reviewer argues that the use case is unclear and that a baseline that uses all samples is missing. In fact the baseline already appears and the use case actually seems very clear from the abstract. For these reasons, the authors have convinced me the paper is worth accepting. Authors: you've already indicated that you've updated the submission to respond to reviewer changes, if you could double check their comments for any recommendation you may have missed on accident that would be great! The paper will make a great contribution to the conference!

**Note From Pc:**

if the above contains the word "oral" or "spotlight" please see: "oral" presentation means -> notable-top-5% and "spotlight" means -> notable-top-25%. As stated in our emails, we are disassociating presentation type from AC recommendations